# Joint multi-ancestry and admixed GWAS reveals the complex genetics behind human cranial vault shape

Seppe Goovaerts [1,2] ✉, Hanne Hoskens[2,3], Ryan J. Eller[4], Noah Herrick[4], Anthony M. Musolf[5], Cristina M. Justice[6,7], Meng Yuan [1,2,3], Sahin Naqvi [8,9], Myoung Keun Lee[10], Dirk Vandermeulen[2,3], Heather L. Szabo-Rogers[11], Paul A. Romitti [12], Simeon A. Boyadjiev[13], Mary L. Marazita [10,14], John R. Shaffer [10,14], Mark D. Shriver[15], Joanna Wysocka [8,16,17], Susan Walsh[4], Seth M. Weinberg [10,14,18] ✉ & Peter Claes [1,2,3,19] ✉

The cranial vault in humans is highly variable, clinically relevant, and heritable, yet its genetic architecture remains poorly understood. Here, we conduct a joint multi-ancestry and admixed multivariate genome-wide association study on 3D cranial vault shape extracted from magnetic resonance images of 6772 children from the ABCD study cohort yielding 30 genome-wide significant loci. Follow-up analyses indicate that these loci overlap with genomic risk loci for sagittal craniosynostosis, show elevated activity cranial neural crest cells, are enriched for processes related to skeletal development, and are shared with the face and brain. We present supporting evidence of regional localization for several of the identified genes based on expression patterns in the cranial vault bones of E15.5 mice. Overall, our study provides a comprehensive overview of the genetics underlying normal-range cranial vault shape and its relevance for understanding modern human craniofacial diversity and the etiology of congenital malformations.

The cranial vault—the globular portion of the head, shaped by flat, plate-like bones that surround and protect the brain—shows considerable size and shape variation within and among human populations[1,2]. Because cranial vault morphology has implications for paleoanthropology[3,4], forensics[5,6], and human health[7–9], it is crucial to understand the factors that drive its phenotypic variation. The debate surrounding the relative contribution of genetic and environmental influences on the cranial vault has a long history, starting with the

[1]Department of Human Genetics, KU Leuven, Leuven, Belgium. [2]Medical Imaging Research Center, University Hospitals Leuven, Leuven, Belgium. [3]Department of Electrical Engineering, ESAT/PSI, KU Leuven, Leuven, Belgium. [4]Department of Biology, Indiana University Indianapolis, Indianapolis, IN, USA. [5]Statistical Genetics Section, Computational and Statistical Genomics Branch, NHGRI, NIH, MD, Baltimore, USA. [6]Genometrics Section, Computational and Statistical Genomics Branch, Division of Intramural Research, NHGRI, NIH, Baltimore, MD, USA. [7]Neurobehavioral Clinical Research Section, Social and Behavioral Research Branch, National Human Genome Research Institute, National Institutes of Health, Bethesda, MD, USA. [8]Department of Chemical and Systems Biology, Stanford University School of Medicine, Stanford, CA, USA. [9]Departments of Genetics and Biology, Stanford University School of Medicine, Stanford, CA, USA. [10]Department of Oral and Craniofacial Sciences, Center for Craniofacial and Dental Genetics, University of Pittsburgh, Pittsburgh, PA, USA. [11]Department of Anatomy, Physiology and Pharmacology, University of Saskatchewan, Saskatchewan, Canada. [12]Department of Epidemiology, College of Public Health, The University of Iowa, Iowa City, IA, USA. [13]Department of Pediatrics, University of California Davis, Sacramento, CA, USA. [14]Department of Human Genetics, University of Pittsburgh, Pittsburgh, PA, USA. [15]Department of Anthropology, Pennsylvania State University, State College, PA, USA. [16]Department of Developmental Biology, Stanford University School of Medicine, Stanford, CA, USA. [17]Howard Hughes Medical Institute, Stanford University School of Medicine, Stanford, CA, USA. [18]Department of Anthropology, University of Pittsburgh, Pittsburgh, PA, USA. [19]Murdoch Children's Research Institute, Melbourne, VIC, Australia. ✉e-mail: seppe.goovaerts@kuleuven.be; smwst46@pitt.edu; peter.claes@kuleuven.be

observation by Boas in the early 20th century[10] that head dimensions can change in a single generation in response to environmental conditions. Formal heritability studies, including those reanalyzing Boas's data[11], indicate that genetic effects account for a sizable portion (>50%) of the phenotypic variation in vault size. Of course, these positions are not in conflict, as continuous morphological traits are generally considered polygenic, with epistatic and gene-environment interactions playing an important role.

Revealing the genetic architecture of the cranial vault is a necessary step toward elucidating the various molecular pathways involved in both normal and abnormal cranial development and growth. However, despite evidence of a genetic contribution, we know little about the specific genes that impact variation in human vault morphology. Several lines of evidence point to signaling molecules, like *Fibroblast Growth Factors* and their receptors, as being important. Genes in several signaling pathways (e.g., FGF, TGF, BMP, WNT, IHH/SHH, TWIST) have been implicated in congenital malformations characterized by vault dysmorphology, such as syndromic and non-syndromic forms of craniosynostosis[12,13]. Genome-wide association studies (GWASs) of non-syndromic craniosynostosis have yielded candidates such as *BMP2*, *BBS9*, and *BMP7*[14,15]. Moreover, gene expression and experimental studies of the developing vault have also implicated many of these genes (and others in their pathways) in suture morphogenesis[16–19]. When considering normal-range variation in cranial vault morphology, genome-wide QTL studies of skull shape in mice have implicated a few dozen genes, including some with effects on the vault[20–22]. In humans, two candidate gene studies have reported associations between common variants in *FGFR1* and cranial vault dimensions[23,24], and several large GWAS of global vault size (head circumference) have identified a handful of loci[25–27]. In addition, a GWAS of vault length, width, and cephalic index in over 4000 individuals reported associations at several loci near relevant genes like *SOX9* and *SOX11*[28]; notably, no association with *FGFR1* was observed.

One of the limitations of prior genetic studies of human vault morphology is a reliance on relatively simple phenotyping approaches (e.g., distances). Such measures are often straightforward to acquire but suffer from an inability to adequately describe complex 3D shapes and may not capture the most salient aspects of variation for genetic investigations. As a result, it is likely that the genes identified to date account for a small fraction of the heritable variation in human vault morphology. In genetic studies of human facial morphology, we have previously shown that data-driven multivariate approaches capable of more fully exploiting the information contained in 3D biological shapes outperform more traditional morphometric approaches[29]. In the present study, we advance the pace of genetic discovery by applying a similar phenotyping strategy to the cranial vault. We accomplish this by extracting 3D vault surfaces from magnetic resonance images (MRIs) collected on a multi-ancestry adolescent cohort, partitioning the surfaces into anatomical regions in a global-to-local pattern, quantifying the shape variation present in each region, and then performing multivariate GWASs. In addition, we test whether our discovered variants impact risk for single suture craniosynostosis, and, given the close relationship between brain, facial, and vault morphology[30], we investigate the degree of overlap between their genetic architectures.

## Results

### Joint multi-ancestry and admixed GWAS of cranial vault shape

Cranial vault shape, herein defined as the outer head surface encompassing the supraorbital ridge and extending towards the occipital bone, was extracted from structural MRIs (Supplementary Fig. 1; Methods). Since the outer soft-tissue layer in this region is thin and uniform[31], shape variation associated with the neurocranial bones is well captured by our phenotypic definition. To study shape variation at both a global and local resolution, the cranial vault surface, represented by a mesh of 11,410 vertices, was partitioned into a set of smaller segments through hierarchical spectral clustering following

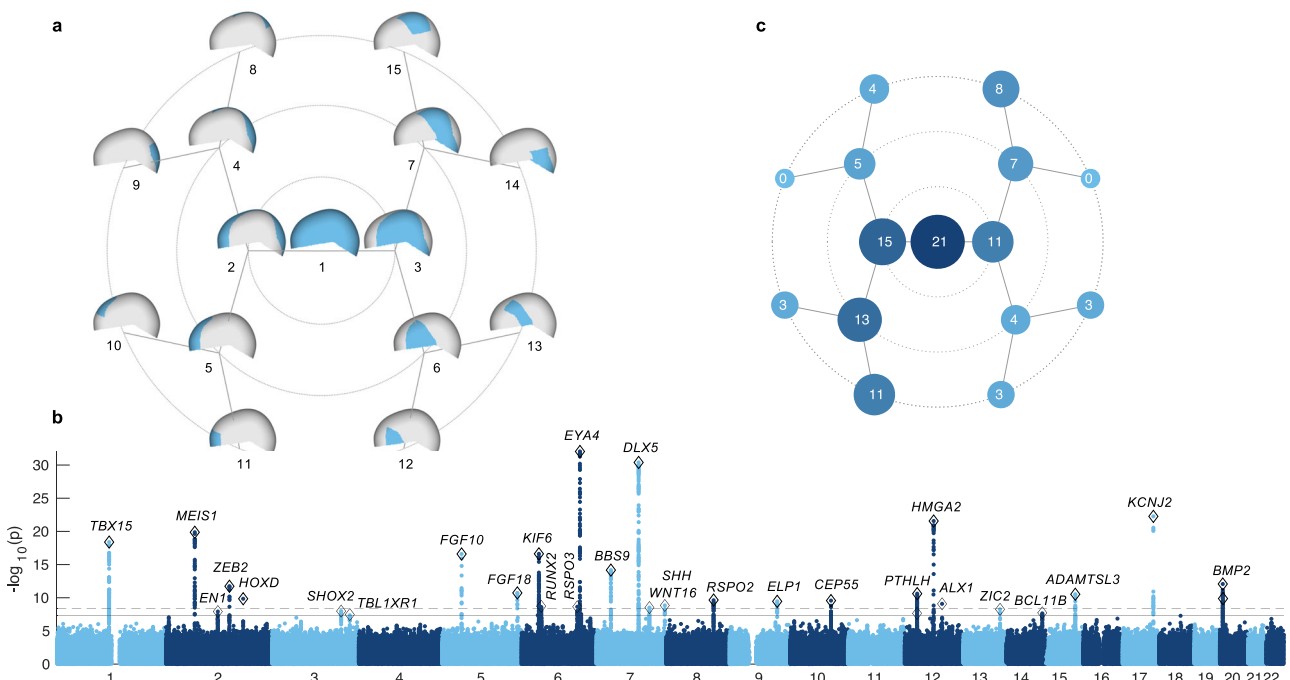

**Fig. 1 | Global-to-local genome-wide association study of cranial vault shape.**
**a** Hierarchical segmentation of cranial vault shape resulting in 15 cranial vault segments (cyan) across 4 hierarchical levels. **b** Manhattan plot of genome-wide associations. For each SNP the lowest *P*-value (CCA, upper-tail chi squared) across all 15 cranial vault segments was plotted. Full and dashed horizontal lines represent genome-wide (*P* < 5e−8) and study-wide (*P* < 4.37e−9) significance thresholds respectively. Plausible candidate genes are annotated to each genome-wide significant locus (*n* = 30) (Methods). **c** Number of genetic loci reaching genome-wide significance (*P* < 5e−8; CCA, upper-tail chi squared) in each cranial vault segment.

the data-driven approach introduced by Claes et al.[29] (Fig. 1a). The result was a set of 15 cranial vault segments, with the first segment being the 'global' full cranial vault surface and segments 8 to 15 being the most 'local' segments. Considering the globular and relatively smooth nature of the cranial vault, in contrast to the face[32] or brain[33], we stopped at the fourth hierarchical level to not unnecessarily inflate the total number of segments. The smallest segment, segment 12, covered 7.70% of the overall surface area.

Each of the 15 cranial vault segments was subject to principal component analysis (PCA) to extract a smaller number of morphological dimensions, followed by parallel analysis to identify the optimal number of PCs to retain. We then applied canonical correlation analysis (CCA), which extracts the linear combination of those dimensions that maximally correlates with the state of a given single nucleotide polymorphism (SNP). In total, 10,647,531 SNPs across the genome were tested in a US cohort comprising of rich ancestral diversity. After applying quality control procedures on images, genotype, and covariate data, the final GWAS cohort consisted of 6772 unrelated individuals.

Given the high level of admixture in our dataset and the lack of more than a single substantial homogenous group, any stratification-based approach, such as a GWAS meta-analysis, was deemed unfruitful and we instead opted to run a single joint multi-ancestry and admixed GWAS. As a compromise between sample inclusion and model complexity, our GWAS was limited to three ancestral groups (i.e., European, African, and Indigenous American), and any admixture thereof. Prior to GWAS, cranial vault shape was adjusted for the effects of age, sex, height, weight, cranial size, MRI machine/scanning site, and ancestral heterogeneity using partial least squares regression (PLSR). The latter was done by including global genomic ancestry, expressed by the first 10 genomic PCs, as well as local genomic ancestry, expressed by ancestral dosages at 0.2 cM windows throughout the genome (Methods).

In total, 15 GWASs were conducted (one per vault segment), yielding 1658 SNPs that reached genome-wide significance ($P < 5e−8$) in at least one of the segments. Of these SNPs 1138 were also study-wide significant ($P < 4.37e−9$), correcting for the effective number of GWAS runs as estimated by permutation testing (Methods). Based on genomic distance and linkage disequilibrium (LD), SNPs were clumped into 30 independent genome-wide and 24 study-wide significant genomic loci (Fig. 1b; Supplementary Table 1). Among these signals, only the locus near *HMGA2* has been previously identified in GWAS on cranial vault dimensions[25,26]. The 30 lead SNPs combined explained 1.03% to 1.83% of variation among the 15 cranial vault segments, and 1.31% of global cranial vault shape variation after adjustment for covariates. We observed a range of associated phenotypic effects, with some GWAS signals impacting multiple regions of the vault simultaneously and others impacting vault shape in a more localized manner (Supplementary Data 1). Animations of the shape effects associated with each locus are available from our FigShare repository[34]. In addition to global vault shape (segment 1), segments involving the frontal portion of the vault showed the largest number of significant associations (Fig. 1c). Twenty-one out of 30 loci (21/30, 70.0%) showed significant ($P < 5e−8$) effects on global cranial vault shape (Fig. 1c and Supplementary Fig. 2), with 13 loci (13/21, 61.9%) providing the most significant $P$-values. Among the 7 loci (7/30, 23.2%) that had most significant effects at the finest level of segmentation (i.e., hierarchical level 4), only 3 (3/7, 42.9%) were also significant ($P < 5e−8$) in the global cranial vault. For most loci with their lowest $P$-values in the global cranial vault, significance gradually decreased as the vault was partitioned into smaller segments. In contrast, most of the loci with their lowest $P$-values in one of the more local level-4 segments exhibited the opposite effect. This suggests a distinction between globally and locally acting loci.

To demonstrate that our data-driven segmentation approach is capturing genetically salient aspects of vault shape variation, we examined the effects of discovered SNPs per vertex on the entire dense 3D mesh. Based on the 21 lead SNPs that were significantly ($P < 5e−8$) associated with global vault shape, we calculated deformation hotspots (Supplementary Fig. 3), which we defined as local regions of the cranial vault where effects attained a greater magnitude relative to their immediate surroundings. Most of these hotspots were located along the midline, coinciding with sutures, and two hotspots were located laterally. These results show that without including any SNPs whose phenotypic associations were dependent on the segmentation approach, a pattern of deformation hotspots was obtained that aligns almost perfectly with our vault segmentation. This provides confirmation that our data-driven approach can capture biologically meaningful information and is capable of doing so with a fraction of the computational bandwidth required to perform vertex-wise analyses.

To test for any uncontrolled confounding due to population stratification, we compared the obtained genome-wide association $P$-values to a set of simulated association $P$-values under the null hypothesis (Methods). No inflation of test statistics was observed (Supplementary Fig. 4), suggesting that our GWAS results were not affected by uncontrolled population stratification. Due to the observed relationship between the fixation index ($F_{ST}$) and power in our GWAS, we calculated the genomic control factor lambda ($\lambda_{GC}$) for different subsets of SNPs, defined by varying upper limits for the $F_{ST}$. The highest $\lambda_{GC}$ of 1.002 was found for the set of SNPs with $F_{ST}$ smaller than 0.001 (Supplementary Fig. 4), further suggesting the absence of uncontrolled confounding.

Next, we replicated our GWAS signals and their phenotypic effects in an independent cohort of 16,846 individuals from the UK Biobank[35], based on MRI. For each of the 30 lead SNPs, we tested all segments in which genome-wide significance was reached ($P < 5e−8$) during the discovery phase, totaling 108 tests (Methods). In total, 55 out of 108 (50.9%) lead-SNP/segment pairs and 20 out of 30 (66.7%) individual lead SNPs were replicated in at least one segment at a 5% false discovery rate (FDR) (Supplementary Table 1). Note that the replication MRIs were heavily damaged, mainly at the frontal area of the cranial vault (i.e., MR images are defaced for subject anonymity). Although these defects were partially mitigated during image processing (Methods), the lowest segment-wise replication rates of 23.1% and 9.09% were observed in the corresponding cranial vault segments 5 and 11, respectively (Supplementary Fig. 5). Thus, our replication rate is likely very conservative.

## Multi-ancestry GWAS signals are shared between ancestries

To investigate whether the lead SNPs representing each of our 30 genome-wide significant loci were shared across different ancestral populations, $F_{ST}$ enrichment analysis was performed following Guo et al. (2018)[36]. Significantly lower inter-African/European, but not inter-Indigenous-American/European $F_{ST}$ was found for our set of lead SNPs compared to the expected values for a set of similar SNPs in terms of minor allele frequency (MAF) and LD score (Fig. 2a, b). Notably, the mean observed inter-African/European (mean: 0.072, range: 0 to 0.320) and inter-Indigenous-American/European (mean: 0.079, range: 0 to 0.195) $F_{ST}$ were similar. However, in line with existing knowledge, the expected inter-African/European $F_{ST}$ was higher than the inter-Indigenous-American/European $F_{ST}$, ultimately resulting in a significant enrichment of shared signals only between African and European ancestry.

To examine how the multi-ancestry aspects of our study influenced the overrepresentation of shared loci, we repeated the $F_{ST}$ enrichment analysis under two additional scenarios. In the first scenario, the multi-ancestry GWAS ($n = 6772$) was rerun, this time without

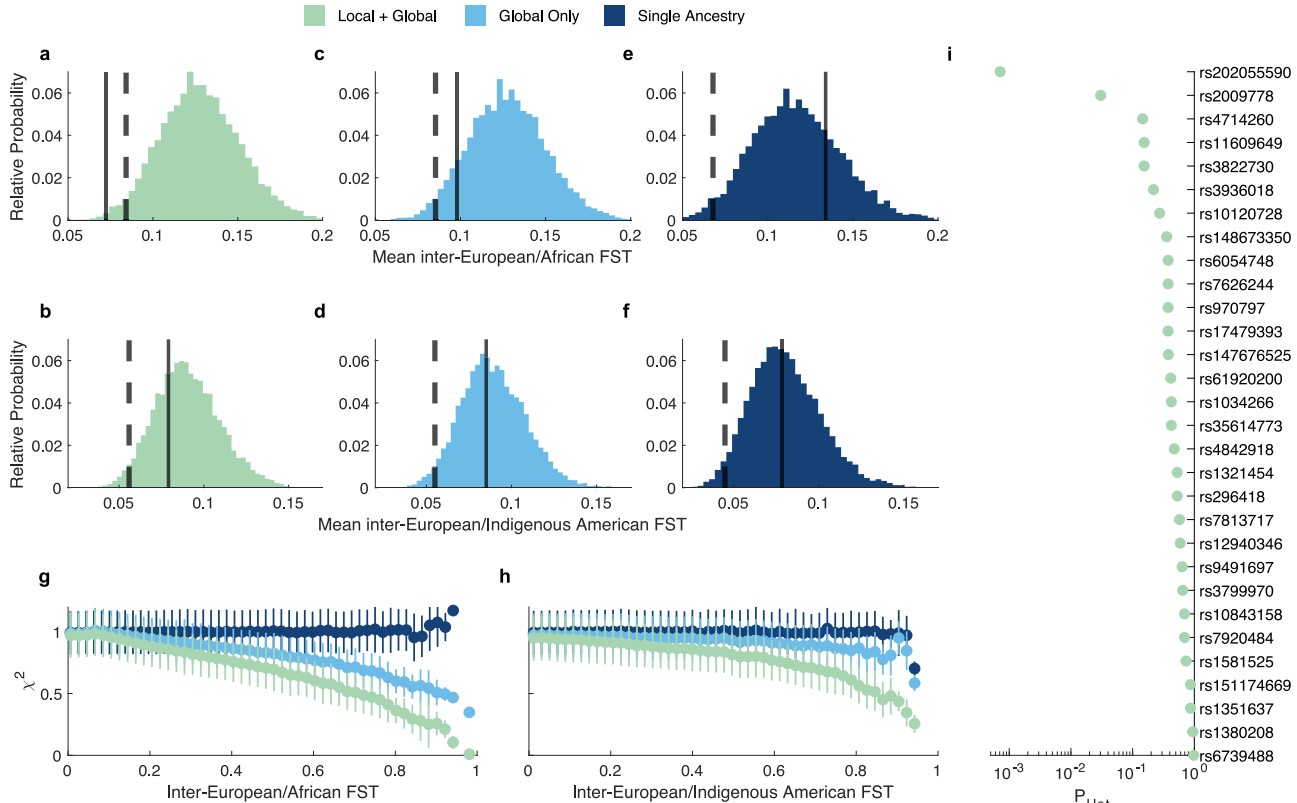

**Fig. 2 | Adjustment for global and local genomic ancestry induces prioritization of shared signals between ancestral groups. a–f** $F_{ST}$ enrichment analysis (top row: European vs. African; bottom row: European vs. Indigenous American) of lead SNPs at genome-wide significant GWAS loci under different scenarios. Left (green, *Local + Global*): the main, multi-ancestry GWAS ($n = 6772$) with global and local genomic ancestry adjustment. Middle (cyan, *Global only*): a multi-ancestry GWAS on the same individuals ($n = 6772$) with only global genomic ancestry adjustment. Right (navy, *Single Ancestry*): a European-only GWAS ($n = 4198$) with global genomic ancestry adjustment. The full vertical line represents the mean observed $F_{ST}$ and the dashed vertical line represents the 2.5th percentile on the distribution of expected $F_{ST}$. **g, h** Mean test statistic of the full cranial vault GWAS (segment 1) across 50 $F_{ST}$ bins (left: European vs. African; right: European vs. Indigenous American) under the GWAS scenarios from **a–f**. Error bars represent the standard deviation of the test statistic. **i** Heterogeneity of effect size between European and African ancestry based on the univariate latent phenotypes associated with each SNP in the main GWAS (Methods). All segment-SNP combinations with $P < 5e−8$ during GWAS discovery were considered, and the lowest $P$-value (upper tail chi-squared test with varying degrees of freedom; see Methods) for each SNP was kept after adjustment for 5% FDR. The vertical line indicates the $P_{adjusted} < 0.05$ threshold.

adjusting for the effects of local genomic ancestry. Comparing the outcome with the main GWAS revealed that the adjustment for local genomic ancestry induces a stronger selection for shared loci between ancestral groups compared to the adjustment for global genomic ancestry alone (Fig. 2c, d). In the second scenario, the GWAS was rerun on only the subjects that were assigned European ancestry ($n = 4198$) (Methods). In this more ancestrally homogeneous group, adjustment for the first 10 genomic PCs was still performed to adjust for the within-Europe population structure. In contrast to the main GWAS, no significant over- or underrepresentation of SNPs shared between ancestral groups was observed (Fig. 2e, f), suggesting that the multi-ancestry nature of our GWAS drove the overrepresentation of shared SNPs.

Figure 2g, h illustrates how the selection of shared loci was achieved through the reduction in statistical power for differentiated SNPs. While, in our multi-ancestry GWAS, adjustment for global ancestry alone reduced the power of high $F_{ST}$ SNPs, the additional adjustment for local genomic ancestry exacerbated this effect. These results align with previous studies[37–39]. Moreover, for any SNP with $F_{ST}$ equal to 1 (i.e., both ancestral groups have a different allele that is fixed within that group) the allelic dosage would correlate perfectly with the local genomic ancestry. Therefore, those SNPs would attain no power after removing the effects of local genomic ancestry (Fig. 2g, h). Together, these results show that our multi-ancestry GWAS approach identified a higher-than-expected sharing of genetic factors underlying

cranial vault shape between ancestral groups. They also illustrate how the selection of shared SNPs between ancestries directly results from our multi-ancestry approach to GWAS.

To further validate the sharing of effects across populations, we tested for heterogeneity of effect size between ancestral groups at each of our 30 lead SNPs under several scenarios (Methods). Briefly, each test is based on the likelihood ratio of nested models, where the full model includes differential effect sizes between ancestral groups, and a constraint model includes a single joint effect size. All testing scenarios yielded mostly consistent results (Supplementary Fig. 6). At 5% FDR, we found heterogeneous effect sizes for 2 out of 30 lead SNPs under the most sensitive testing scenario (Fig. 2i) and for 3 out of 30 lead SNPs across all scenarios. This lack of significant effect size heterogeneity further supports that the identified loci comprise shared signals among ancestral groups. Notably, no significant relationship was observed between $F_{ST}$ and effect size heterogeneity (Supplementary Fig. 7).

We next performed a sensitivity analysis to evaluate the discovery of genome-wide loci when including or excluding the 2504 non-European-ancestry GWAS cohort subjects. To this end, a locus was considered to be overlapping between both GWASs if the corresponding lead SNPs were within 250 kb, resulting in a joint set of 31 unique loci (Supplementary Fig. 8). Among these loci, 22 were shared, of which 17 (17/22; 77.3%) reached lower $P$-values in the multi-ancestry GWAS (Supplementary Fig. 8). Moreover, 8 loci were unique to the

multi-ancestry GWAS, and one locus (*rs563186113*) was unique to the European-only GWAS. Nonetheless, all those 9 loci stilled showed some degree of association in the other GWAS. Overall, the multi-ancestry GWAS thus achieved a higher discovery rate, identifying 7 more loci, and reaching lower *P*-values in 25 of the 31 loci (80.6%). Furthermore, shape effects and segment-wise association profiles were highly concordant between both GWASs at each of the 31 loci (Supplementary Data 2). At 21 loci (21/31; 67.7%), the same segment generated the lowest *P*-value, and at 5 more loci (5/31; 16.1%) the lowest *P*-values were generated in segments that were directly related. Strong co-localization was observed for the genomic signals in both GWASs as shown by LocusZoom[40] and LocusCompare[41] plots in Supplementary Data 2. Altogether, the analyses conducted in this section extensively demonstrated which aspects of the genetic findings were influenced by the inclusion of non-European individuals.

## Cranial vault shape originates early in development
Using the genomic regions enrichment of annotations tool (GREAT[42]), genes near the 30 lead SNPs were tested for enrichment of biological processes and pathways, as well as associated mouse phenotypes (Supplementary Table 2 and Supplementary Table 3). Among the enriched (by both binomial and hypergeometric test, 5% FDR) mouse phenotypes were terms related to abnormal head morphology (abnormal morphology of the head, face, mouth, cranium, neurocranium, viscerocranium, basicranium; exencephaly), and abnormal morphology of multiple craniofacial bones (abnormal morphology of the squamosal, interparietal, temporal, occipital, supraoccipital, zygomatic, basisphenoid, nasal bone; maxilla, mandible, palate), as well as other bones throughout the body (abnormal morphology of the humerus, scapula, tibia, clavicle, rib, limb long bone, …). Interestingly, the analysis identified significant terms related to the brain and neural tube (open neural tube, abnormal neural tube closure, decreased midbrain size, abnormal midbrain size, absent cerebellar lobules, abnormal neural crest cell migration). These results illustrate the important role of the genes near our identified loci in normal craniofacial development and suggest an overlap in genetic architecture between the cranial vault, the face, and the brain.

In line with the associated mouse phenotypes, we found enriched biological processes related to bone development (skeletal system development, osteoblast differentiation, bone development, ossification, osteoblast development) and cartilage development (regulation of cartilage development, chondrocyte differentiation, cartilage development, positive regulation of cartilage development). Furthermore, the analysis yielded enrichments for mesenchyme-related terms (mesenchymal cell differentiation, mesenchyme development, connective tissue development), as well as more broad terms related to embryonic development (embryo development, embryonic morphogenesis, embryonic organ morphogenesis and development, …). In agreement with these terms, enrichment of phenotypes in GWAS Catalog using FUMA yielded bone-related traits, including, bone mineral density (heel, skull, lumbar spine, …) and height (Supplementary Table 4).

Next, we analyzed H3K27ac ChIP-seq signals near the 30 lead SNPs across approximately 100 cell types and tissues (Fig. 3). Additional information regarding the cell types and tissues can be found in Supplementary Table 5. We found that the 30 GWAS loci were most enriched for nearby H3K27ac signal in embryonic craniofacial tissues and cranial neural crest cells (CNCCs), indicative of cell-type specific enhancer activity. This suggests that our GWAS signals are located near enhancer elements that are active during early craniofacial development. Notably, while the frontal bone originates from CNCCs, the parietal bone arises from the mesoderm-derived mesenchymal progenitors. However, the H3K27ac ChIP-seq data from the latter mesenchymal progenitors were not available, and thus were not included in this analysis. Nonetheless, the majority of the identified GWAS loci affected either global cranial vault shape or the frontal segments, consistent with the strong contribution of CNCC-derived structures. Together, these results suggest that post-natal shape variation associated with the neurocranial bones originates in early developmental stages.

## Expression of GWAS candidate genes in E15.5 mice calvarium
To compare the parietal versus frontal localization of phenotypic effects associated with our lead SNPs to expression levels of associated

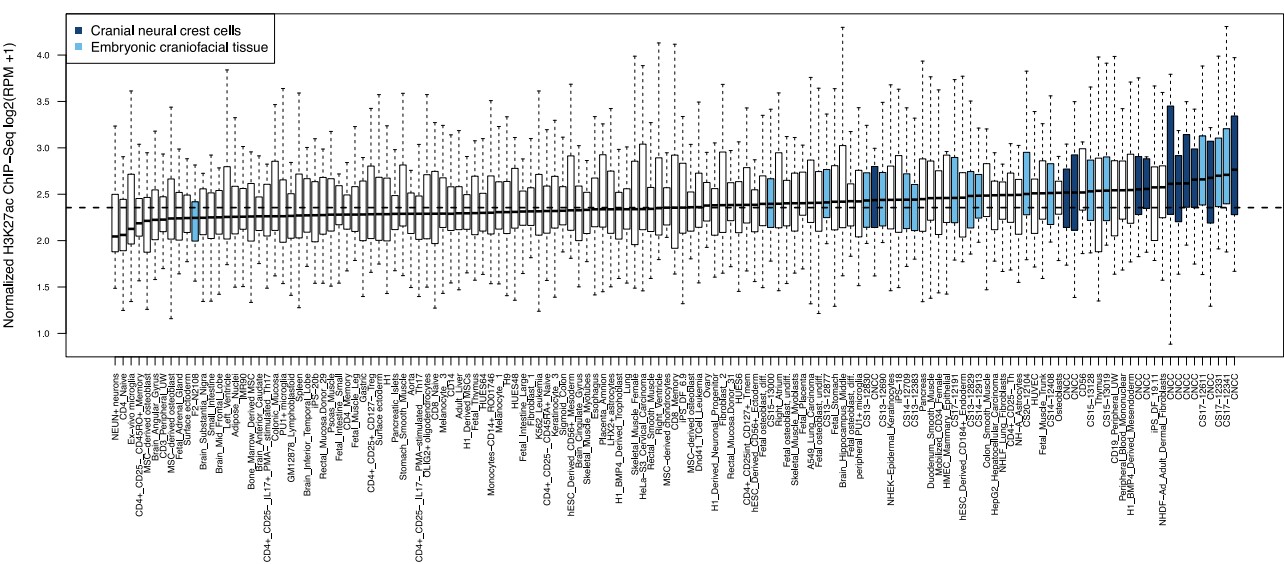

**Fig. 3 | Regions near the 30 genome-wide significant lead SNPs are enriched for active enhancers with preferential activity in cranial neural crest cells and embryonic craniofacial tissue.** Enrichment for active enhancers in cranial neural crest cells and embryonic craniofacial tissues was significant (*P*-value = 1.37e−10, Wilcoxon rank sum test, one-tailed, *n* = 141 cell/tissue types). Each boxplot (*n* = 30 loci) represents the distribution of H3K27ac signal in 20-kb regions around the 30 genome-wide-significant (*P* < 5e−8) lead SNPs in one sample, with cranial neural crest cells (navy) and embryonic craniofacial tissues (cyan) highlighted. Boxplots plot the first and third quartiles, with a dark black line representing the median. Whiskers extend to the largest and smallest values no further than 1.5× the interquartile range from the first and third quartiles, respectively. The horizontal dashed line represents the median level of H3K27ac reads per million (RPM) signal across all cell types and tissues.

candidate genes, we performed a differential expression analysis of E15.5 mice parietal and frontal bone tissues. We found 648 differentially expressed genes (DEGs) out of approximately 30,000 analyzed genes from the sequencing library. Of the DEGs, 410 were upregulated in the parietal bone tissues and 238 were upregulated in the frontal bone tissues (Supplementary Table 6). We were particularly interested in exploring the candidate genes (Supplementary Table 1; Methods) located near the 30 GWAS lead SNPs to determine whether a) the candidate genes' mouse homologs demonstrated differential expression and b) given the genes were differentially expressed, whether their expression patterns could further validate the cranial vault phenotypes in which the associated lead SNPs were initially found. Among the GWAS candidate genes were 11 DEGs at 5% FDR, of which nine genes showed significantly higher expression in the frontal bone, and two in the parietal bone (Table 1 and Supplementary Fig. 9). Other candidate genes were either not differentially expressed or had fewer than one raw (unnormalized) gene count in at least one of the six samples.

To compare between the frontal and parietal phenotypes utilized in the expression analysis and the 15 cranial vault segments used in the GWAS, all GWAS lead SNPs were assigned a 'frontal', 'parietal', or 'both' label, depending on the association profile across the 15 segments (Methods). As GWAS lead SNPs were identified using vault segments from several overlapping hierarchical layers, we performed our parietal/frontal/both classifications twice using the following two schemes—'most significant hit' and 'most specific hit'. Both classifications schemes yielded concordant results, demonstrating robustness of the assigned labels.

For example, GWAS lead SNP rs11609649, associated with *ALX1* via GREAT, was identified in cranial vault segment 5 in the GWAS analysis, which corresponds to the frontal region. Since segment 5 contains the most significant signal for the lead SNP and since segment 5 is also phenotypically specific, the classification of *ALX1* for the 'most significant hit' and 'most specific hit' method is the same—frontal. For several other loci, the most significant signal was found in a segment with both parietal and frontal bone content, such as the full cranial vault (segment 1). Nonetheless, their association may still be driven by a predominant frontal or parietal activity. For those loci, we identified the most specific segment by tracing the association signal into hierarchical levels three and four.

For the majority of DEGs, the GWAS analysis identified a stronger effect in the region where the gene was upregulated in mice. Among those genes were *Alx1, Eya4, Hmga2, Shox2, Cped1,* and *En1*, which all showed higher transcriptional activity in the frontal bone and *Rspo3*

and *Bmper*, which showed higher transcriptional activity in the parietal bone. Conversely, while *Fgf10, Zeb2,* and *Adamtsl3* were downregulated in the murine parietal bone, the GWAS analysis identified a stronger effect in the parietal region. Specifically, the association with rs3822730 near *FGF10* was most significant in a parietal segment (segment 8; Supplementary Data 1) and genome-wide significant in other parietal segments (segments 4, 7, and 15) while not reaching nominal significance in any of the frontal segments (segments 5, 10, 11, and 12). The strong parietal localization of the *FGF10* GWAS signal combined with a lower transcriptional activity may indicate differential FGF10 dosage sensitivity between both tissues, where the developing parietal bone could be more sensitive to small changes in FGF10 levels. Alternative explanations exist and exact mechanisms could be a subject for functional follow-up studies. Altogether, our data suggests that some genes may have a predominant frontal or parietal contribution to cranial vault shape.

### Cranial vault shape genetics comprise risk for NCS

Craniosynostosis is a condition that occurs in approximately 1 in 2500 newborns and is characterized by the premature fusion of one or more cranial sutures, thereby drastically affecting skull morphology during growth. The most common form, non-syndromic craniosynostosis (NCS), is etiologically complex and influenced by genetic and environmental factors. A previous GWAS[14] implicated SNPs near *BMP2* and *BBS9* as genetic risk loci for developing sagittal NCS. Interestingly, both genes were also identified in the current GWAS of normal-range cranial vault shape. We tested whether the lead SNPs from both GWASs tagged the same genomic loci. Published risk SNPs for sagittal NCS, rs10262453 ($r^2$ with rs148673350: 0.98; 1000 G all populations) near *BBS9* and rs1884302 ($r^2$ with rs6054748: 0.95; 1000 G all populations) near *BMP2* reached *P*-values of 3.02e−14 and 2.40e−10 respectively in our normal-range cranial vault shape GWAS, thus both reaching study-wide significance (Fig. 4a, b and Supplementary Fig. 10). Remarkably, the latent shape variation associated with our lead SNP near *BMP2*, rs6054748, comprised an elongation and narrowing of the cranial vault and when exaggerated, presented a dolichocephalic cranial vault shape, reminiscent of sagittal NCS patients (Fig. 4c–e). No such resemblance was observed for our *BBS9* lead SNP (Supplementary Data 1).

To further assess whether any other genomic loci involved in normal-range cranial vault shape were contributing risk for developing sagittal NCS, we proceeded with a targeted replication of our 30 GWAS loci in a sagittal NCS cohort consisting of 63 case-parent trios ($n = 189$). At 5% FDR, significant associations were identified for our lead SNPs

**Table 1 | Classification of frontal versus parietal localization based on global-to-local GWASs and RNAseq in murine (E15.5) parietal and frontal bones**

| GWAS SNP | Mouse gene | Log₂-fold change | Differential expression $P_{adjusted}$ | RNAseq Label | GWAS Label (most specific hit) | GWAS Label (most significant hit) |
|---|---|---|---|---|---|---|
| rs11609649 | *Alx1* | −1.37 | 9.52e-13 | Frontal | Frontal | Frontal |
| rs296418 | *Eya4* | −1.90 | 2.57e-17 | Frontal | Frontal | Frontal |
| rs151174669 | *Hmga2* | −0.48 | 3.63e-2 | Frontal | Frontal | Frontal |
| rs7626244 | *Shox2* | −1.96 | 2.25e-17 | Frontal | Frontal | Frontal |
| rs1581525 | *Cped1* | −0.34 | 9.08e-3 | Frontal | Frontal | Frontal |
| rs6739488 | *En1* | −0.61 | 4.52e-7 | Frontal | Frontal | Both |
| rs4842918 | *Adamtsl3* | −1.01 | 3.18e-12 | Frontal | Parietal | Both |
| rs17479393 | *Zeb2* | −0.34 | 2.81e-2 | Frontal | Parietal | Both |
| rs3822730 | *Fgf10* | −0.52 | 1.61e-2 | Frontal | Parietal | Parietal |
| rs148673350 | *Bmper* | 0.68 | 6.85e−-7 | Parietal | Parietal | Both |
| rs9491697 | *Rspo3* | 0.44 | 6.86e-3 | Parietal | Parietal | Parietal |

All candidate genes near the 30 genome-wide significant lead SNPs were tested for differential expression between the murine cranial tissues. Only significantly differentially expressed genes are shown. *P*-values (Wald test, one-tailed) are adjusted for 5% FDR, log₂-fold change represents the expression levels in the parietal bone relative to the frontal bone.

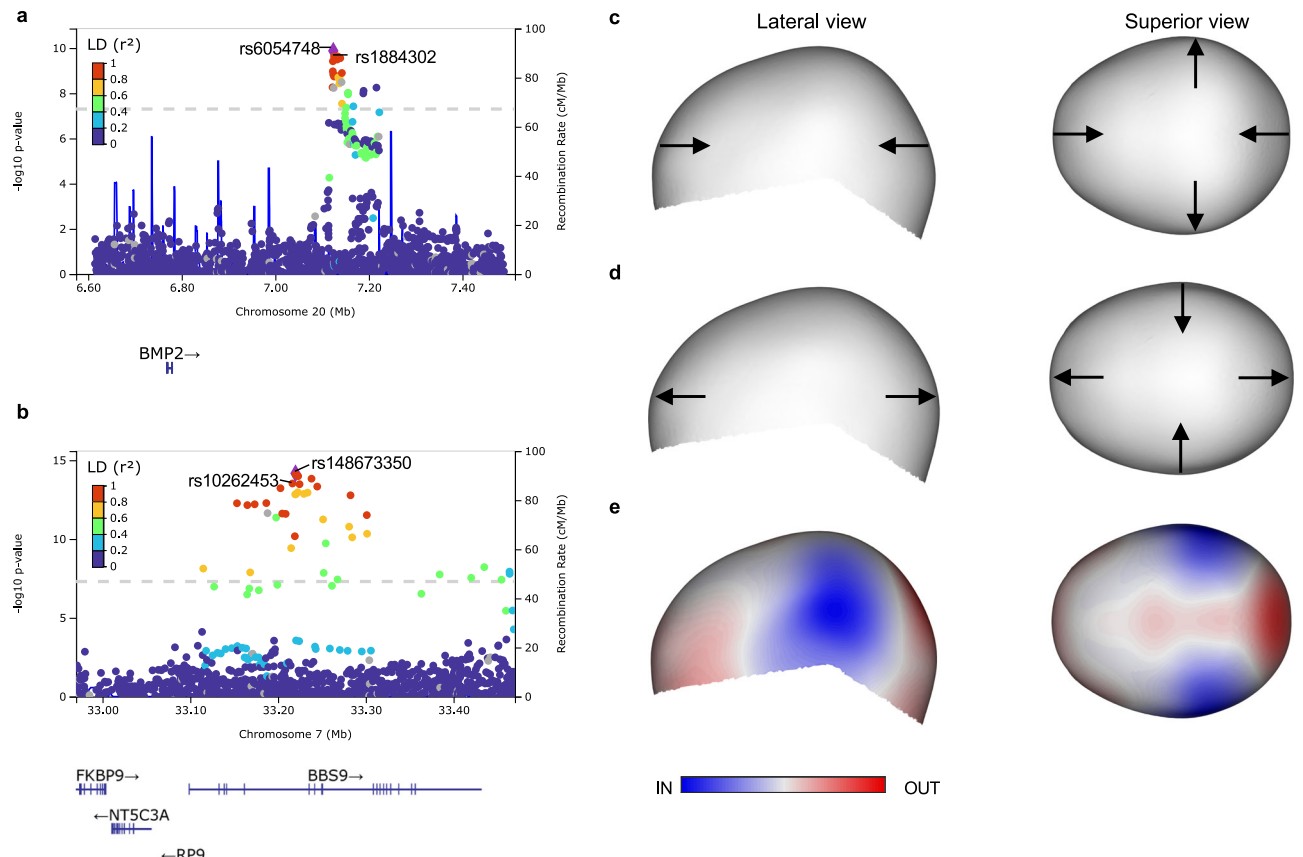

**Fig. 4 | Variation near *BMP2* and *BBS9* comprises risk for sagittal nonsyndromic craniosynostosis. a** LocusZoom plot around rs6054748 near *BMP2*. *P*-values (CCA, upper-tail chi squared) are show for cranial vault segment 4 where rs6054748 was most significant. Sagittal NCS risk SNP rs1884302 from Justice et al. (2012) is also indicated. **b** LocusZoom plot around rs148673350 intronic in *BBS9*. *P*-values (CCA, upper-tail chi squared) are show for cranial vault segment 1 where rs148673350 was most significant. Sagittal NCS risk SNP rs10262453 from Justice et al. (2012) is also indicated. The color in **a** and **b** indicates the LD ($r^2$) with the lead SNP (purple diamond). **c**, **d** Exaggerated depiction of the latent cranial vault shape associated with both alleles of rs6054748 near *BMP2*. Arrows indicate the direction of deformation with regard to the mean cranial vault shape. **e** Latent cranial vault shape associated with rs6054748 near *BMP2*, visualized on the mean cranial vault shape. Blue and red indicate an inward and outward deformation respectively.

rs6054748, rs148673350, and rs1034266 near *BMP2*, *BBS9*, and *ZIC2* respectively. This ultimately resulted in a bidirectional replication for *BMP2* and *BBS9*, as well as an additional candidate risk gene for sagittal NCS, *ZIC2*, not previously reported. Furthermore, we identified significant associations at the nominal *P*-value threshold ($P < 0.05$) with 6 other genomic loci in 92 coronal NCS trios ($n = 276$), 62 metopic NCS trios ($n = 186$), and 17 lambdoid NCS trios ($n = 51$) (Supplementary Table 7). It is also worth noting that rs6054748 near *BMP2* reached at least nominal significance in all but the metopic NCS cohort.

## Shared genetics of the face, brain, and cranial vault

Next, we examined the sharing of genetic signals between the cranial vault and the face. Given the unsigned test statistics yielded by CCA, the approach of calculating genomic correlation using LD-score regression[43], which requires signed summary statistics, was not applicable and we instead used Spearman correlations[33]. Based on a facial GWAS in a European cohort by White & Indencleef et al. (2021)[32] who used a similar global-to-local approach to phenotyping facial shape, comparisons could be made for multiple facial and cranial vault segments. At 5% FDR, significant sharing of genetic signals was observed between the frontal cranial vault segment and all the facial segments, though weaker for the upper lip, nose, and philtrum (Fig. 5a and Supplementary Fig. 11a). Other segments of the cranial vault showed significant, but weaker genetic overlap with the face. These results align with biology, as the face and frontal bone are both derived from CNCCs, but the rest of the cranial vault stems from the

mesoderm[44]. Unsurprisingly, a strong overlap was observed for segments of the face and cranial vault that contained the forehead (Fig. 5a and Supplementary Fig. 11a), illustrating consistency with previously published data[32] and doubling as good validation for those loci affecting forehead morphology.

Based on LD ($r^2 > 0.2$; 1000 G all European populations), we found direct genomic overlap between facial GWAS loci and 18 out of our 30 (60.0%) cranial vault loci (Fig. 5b). Genes near these shared loci have established roles in CNCCs and/or craniofacial skeletal development and comprised a high incidence of involvement in craniofacial syndromes, often with distinct effects on facial appearance, e.g., *TBX15* (Cousin Syndrome, OMIM:260660), *HMGA2* (Silver-Russell Syndrome 5, OMIM:618908), *ZEB2* (Mowat-Wilson Syndrome, OMIM:235730), *ALX1* (Frontonasal Dysplasia 3, OMIM:613456), *MEIS1* (Cleft Palate, Cardiac Defects, And Mental Retardation, OMIM:600987), *EN1* (Endove Syndrome, Limb-brain Type, OMIM:619218), *RUNX2* (Cleidocranial Dysplasia, OMIM:119600), *BMP2* (Short Stature, Facial Dysmorphism, and Skeletal Anomalies with or without Cardiac Anomalies, OMIM:617877) (Supplementary Table 8).

Next, we looked at the genetic correlation between the cranial vault and brain using data from a cortical surface morphology GWAS in a European cohort by Naqvi et al. (2021)[33] who also employed a global-to-local approach to phenotyping. Sharing of genetic signals was observed across the brain and cranial vault segments, with strong sharing between the brain and the frontal segment of the cranial vault, likely relating to their shared ectodermal origin and physical proximity

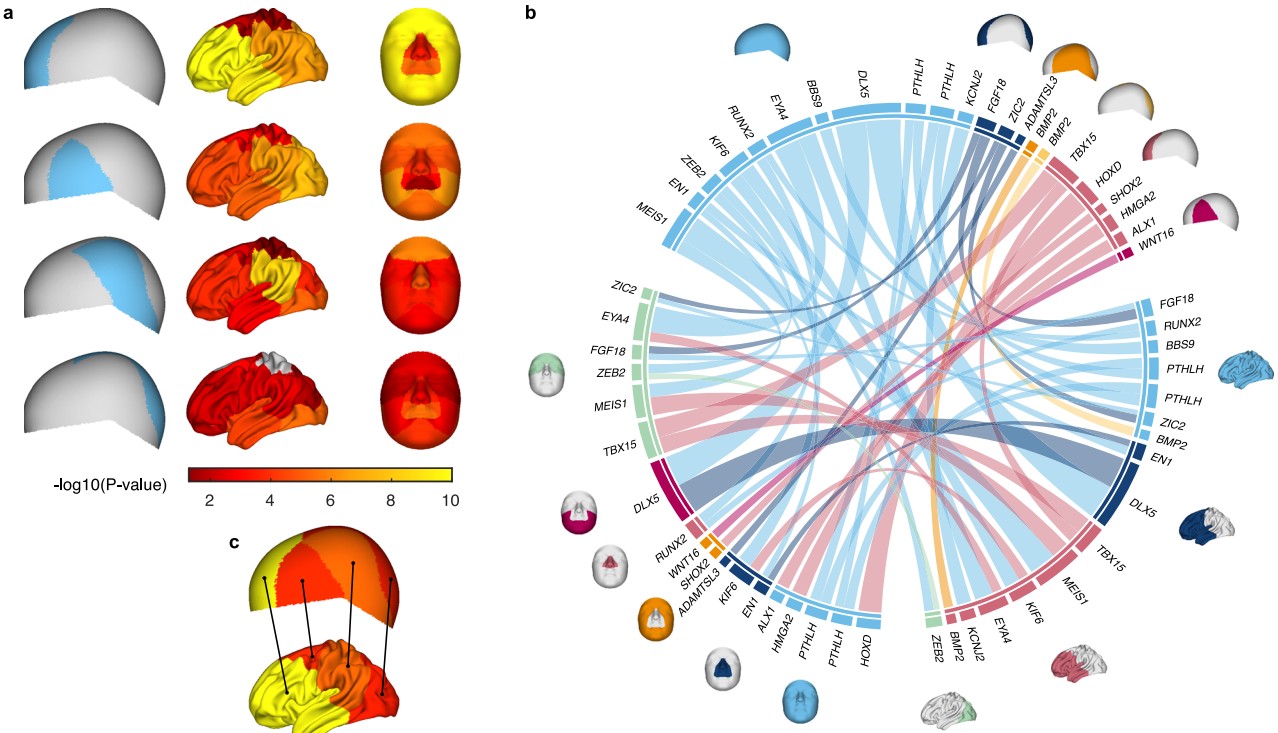

**Fig. 5 | Genome-wide sharing of signals between the cranial vault, brain, and face. a** Genetic Spearman correlations with individual level-3 cranial vault segments (cyan) across level-4 brain and facial segments. Significance of Spearman correlations was determined based on standard errors obtained through bootstrapping. Empirical *P*-values (one-tailed) were adjusted for 5% FDR. Insignificant segments are indicated in grey. **b** Chord diagram of loci shared (r$^2$ > 0.2) between the cranial vault and the brain/face. Line thickness is proportionate to the sum of -log10(*P*-values) (CCA, upper tail chi squared) between both shapes after

normalizing the -log10(*P*-values) per shape so that the maximum value was equal to 1. Loci are grouped by their most significantly associated shape module as indicated by the colors of corresponding circle segments and shape modules. **c** Mutual highest normalized correlations between brain and cranial vault segments indicated by black lines. Segment-wise vault-brain correlations were normalized by the maximum value across the brain and vault respectively. Colors indicate the strength of the unnormalized pair-wise correlations using the same color scale as **a**.

during development (Fig. 5a and Supplementary Fig. 11b)[44]. To further expose patterns of shared genetic signal irrespective of dominant drivers such as shared cellular origins, segment-wise vault-brain correlations were normalized (by dividing by the maximum value across a cranial vault or brain segment). As such, the dominance of the frontal cranial vault segment and potentially others among the brain and cranial vault segments was mitigated. Mutual strongest connections were then extracted, revealing evidence for spatially dependent sharing of genetic signals between the brain and cranial vault (Fig. 5c). These results suggests that the encapsulation of the brain by the cranial vault and the proximities it imposes, is at least in part responsible for their shared genetics.

Sharing of genetic signals between the brain and the cranial vault may be facilitated through mechanical interactions, where intracranial pressure promotes the remodeling of the cranial vault to mirror the shape of the expanding brain[45]. Additionally, brain and skull development may rely on coordinated integration of signaling through fibroblast growth factor (FGF), bone morphogenic proteins (BMP), Wnt, and hedgehog[30]. In the current GWAS, we identified genes encoding members from all four signaling families with established roles in brain development and craniofacial skeletal development, FGF (*FGF10*[46,47], *FGF18*[48,49]), BMP (*BMP2*[50,51]), hedgehog (*SHH*[52,53]), and Wnt (*WNT16*[54,55]). Furthermore, we found that genes near a substantial portion of the 30 cranial vault loci showed evidence of involvement in brain development in mouse knockout studies (Supplementary Table 9). Abnormal brain morphology or size has been reported for mouse knockouts of *MEIS1*[56], *EN1*[57-61], *KIF6*[62], *SHH*[63-68], *HMGA2*[69], *ALX1*[70], *ZIC2*[71-73], and *CEP55*[74]. In addition, several genes have been implicated in the early stages of abnormal neuronal development, tracing back to the neural

plate (*ZEB2*[75], *ZIC2*[76]), neural tube (*EN1*[77], *ZEB2*[75,78,79]), and neural crest (*ZEB2*[78], *ZIC2*[80], *ALX1*[81]).

Among the shared (in LD: r$^2$ > 0.2) brain-face-cranial vault loci (Fig. 5b) included genes with known roles in cranial neural crest development and migration, such as *ZEB2*[78], *ZIC2*[80], *DLX5*[82], and although corresponding lead SNPs were not in strong LD (r$^2$ < 0.2), also *BMP2*[83] and *ALX1*[81]. Other shared loci include craniofacial skeletal development genes, such as *EN1*[84], *FGF18*[49], *BMP2*[50,51], *DLX5*[82,85,86], *RUNX2*[51,85,87–92], *PTHLH*[93], and *TBX15*[94]. In part, this sharing of genetic loci is likely due to pleiotropy, with genes like *ZEB2*, *ZIC2*, and *DLX5* being expressed in both the neural crest and the brain. In other cases, however, genes like *ALX1*, *RUNX2*, and *TBX15* have roles primarily in the mesenchyme with no expression in the brain. Therefore, sharing of these loci is likely driven by the cranial vault.

## Discussion

In this study, we used an unbiased multivariate approach to phenotyping normal-range cranial vault shape, expanding on previous work on the face and brain. By doing so, we have accelerated the pace of discovery by comprehensively documenting the genetics underlying normal-range cranial vault shape variation in humans. Overall, the 30 genome-wide significant loci reflect known aspects of cranial vault biology. Many of the genes near these loci are involved in craniofacial skeletal development, with *RUNX2* being a core transcription factor for other skeletal development genes, and essential for intramembranous ossification, the process through which the cranial flat bones are formed[85,87–93]. Several other candidate genes identified near our GWAS loci encode activators of *RUNX2* transcription, such as *PTHLH* and *DLX5*[85,89,93]. This is a likely pathway through which they exert their

influence on cranial vault shape. In addition, we identified members of the FGF, BMP, hedgehog, and Wnt signaling families, known to modulate different stages in the differentiation from mesenchymal stem cells to osteoblasts[47,49,51,85,88,91–93,95]. For example, SHH has been shown to promote osteoblast differentiation through the activation of *BMP2*, which acts both upstream (through *DLX5*) and downstream of *RUNX2*[88,90,95]. Previous genetic studies on cranial vault dimensions have identified variation near *FGFR1*[23,24], which, together with *FGFR2* and *FGFR3*, is under direct transcriptional control of RUNX2[92]. Though no association was found with *FGFR1* here, we did find associations with *FGF10* and *FGF18*, whose proteins are expressed in the calvarial mesenchyme and have high affinity for FGFR2 and FGFR3 respectively[88,91,92,96]. It has been proposed that increased signaling by FGFs, including FGF10 and FGF18 drives a switch from *FGFR2* to *FGFR1* expression in osteoprogenitor cells, which is associated with the onset of osteogenic differentiation[96–98]. We also identified *EN1*, which was found to regulate proliferation and differentiation of osteoblasts by influencing responsiveness to FGFs through attenuation of *FGFR1* and *FGFR2* expression[84]. Moreover, we identified *WNT16*, which induces osteogenesis and suppresses osteoclast differentiation, cells involved in bone resorption[99].

During normal craniofacial development, a balance between proliferation and differentiation of the suture mesenchyme keeps the sutures patent to accommodate the growing brain[30,100]. Premature ossification of the cranial sutures results in craniosynostosis[16]. Interestingly, several genes identified through our GWAS, or their interacting partners have been implicated in syndromic and non-syndromic craniosynostosis. For example, early expression of *RUNX2* induces ossification in the suture mesenchyme resulting in craniosynostosis[101]. Importantly, gain-of function coding variants in *RUNX2* were reported in patients with midline craniosynostosis and the *RUNX2* p.Ala84-Ala89del variant was reported to be significantly enriched in sagittal NCS, implicating overexpression of this gene in the etiology of craniosynsotosis[102]. We also identified *BCL11B*, a transcription factor involved in keeping suture patency by preventing expression of *RUNX2* and *FGFR2* in the suture mesenchyme[103] and for which de novo mutations have been observed in several craniosynostosis cases[104,105]. In addition, we identified *PTHLH*, whose associated receptor gene, *PTH2R* is implicated in syndromic craniosynostosis[106]. While no significant associations were found near *FGFR1*, *FGFR2*, or *FGFR3*, all implicated in syndromic and non-syndromic craniosynostosis, we found associations with genes encoding their ligands, *FGF10* and *FGF18*[13]. In addition, *RAB23* mutations cause craniosynostosis through failure to repress *FGF10* expression[107], and *ZIC1*, implicated in syndromic craniosynostosis, exerts its pathogenic effect through transcriptional regulation of *EN1*, which we identified in our GWAS[108].

Furthermore, we identified *BMP2* and *BBS9*, which were both implicated as risk genes with large effect sizes (odds ratio > 4 for both loci) in a GWAS on non-syndromic sagittal craniosynostosis[14]. Using data from that GWAS, we performed a cross-GWAS replication of the lead SNPs near *BMP2* and *BBS9* and found bi-directional significant associations for both. Consistent with its effect on sagittal suture morphogenesis, we found our lead SNP near *BMP2* to be associated with an elongation of the cranial vault. These results suggest that some variants affecting normal-range cranial vault shape variation may do so by attenuating the timing of cranial suture closure. In fact, a recent report has concluded that undiagnosed sagittal craniosynostosis is common in the general population, detected in 41 out of the 870 (4.71%) patients based on a retrospective review of head computed tomography scans, but goes unnoticed if the patient has a cranial index within the normal range (width/length > 0.7)[109]. The prevalence of this normocephalic sagittal craniosynostosis increased until 36 months of age, then plateaued, suggesting that, indeed, the timing of cranial suture closure is a source of normal-range cranial vault shape variation with milder features at increased age of synostosis[109].

Though not identified in the current GWAS, several other genes (*ERF*[110], *SIX2*[111], *SMAD6*[111], *SMURF1*[111], *MSX2*[112], *ALX4*[112], *TWIST1*[113], *TCF12*[113], *EPHA4*[114], and *FREM1*[115]) implicated in syndromic and non-syndromic forms of craniosynostosis affect ossification of the suture mesenchyme through modulation of BMP, FGF, or HH signaling and/or attenuation of *RUNX2* expression, highlighting the shared molecular pathways of craniosynostosis and normal cranial vault development. Because of this spatiotemporal overlap, variation at our 30 GWAS loci may work synergistically with rare variants in other relevant genes, modifying their effect. One such example is the suggested interaction between common variation near *BMP2* and rare loss of function mutations in the gene encoding its inhibitor, *SMAD6*[116]. Mouse models of craniosynostosis have supported the idea of modifying genes, whereby mice carrying identical *Fgfr2* mutations exhibited variable phenotype expression depending on their genetic background[117]. Taking into account such interactions may guide surgical interventions and treatment planning in craniosynostosis patients and ultimately lead to better outcomes.

Sharing of loci between the cranial vault and the face and/or the brain was twofold. On the one side, shared loci harbor genes, such as *ZEB2*[78], *ZIC2*[80], *DLX5*[82], *ALX1*[81], and *BMP2*[83], with known roles in the neuroectoderm, which constitutes the shared cellular origin of the brain, facial skeleton, and frontal bone, while the other bones of the cranial vault have different cellular origins, i.e., the mesoderm[44]. In line with the hypothesis that those genes concurrently influence brain, facial and cranial vault shape through their common tissue origins, we detected strong sharing of genetic signals between the frontal segment of the cranial vault and both the brain and face, much stronger than any sharing of signal with the rest of the cranial vault. On the other hand, shared loci comprised a substantial number of genes involved in craniofacial skeletal development: *EN1*[84], *FGF18*[49], *BMP2*[50,51], *DLX5*[82,85,86], *RUNX2*[51,85,87–92], *PTHLH*[93], and *TBX15*[94]. Moreover, mutations in genes with involvement in the cranial neural crest or craniofacial skeletal development result in malformations of the cranial bones, such as in Mowat-Wilson syndrome (*ZEB2*), frontonasal dysplasia 3 (*ALX1*), cleidocranial dysplasia (*RUNX2*), Endove syndrome (*EN1*), cousin syndrome (*TBX15*), and sagittal craniosynostosis (*BMP2*). One explanation could be that these genes promote remodeling of the cranial skeleton to accommodate the growing brain, allowing for subtle changes in brain shape. This is especially plausible given that we detected spatially dependent sharing of genetic signals between the brain and cranial vault. Furthermore, mutations in several genes implicated in our GWAS result in abnormal brain morphology or size in mice, such as *MEIS1*[56], *EN1*[57–61], *KIF6*[62], *SHH*[63–68], *HMGA2*[69], *ALX1*[70], *ZIC2*[71–73], and *CEPS5*[74]. For example, mutations in *KIF6* result in increased brain size, and a domed cranium, which likely results from the force exerted by the brain[62]. This raises the question whether cranium morphology influences brain morphology, or whether it is the other way around. Likely the influence is bidirectional, and starts early in development, as we have shown elevated transcriptional activity near our GWAS loci in CNCCs and embryonic craniofacial tissue. From then, this mutual influence may continue throughout development. Altogether, shared loci between the cranial vault and the face and/or brain comprise genes implicated in diverse molecular pathways across different stages of brain and craniofacial development. The exact mechanisms of action are a relevant topic for future studies.

As we have shown previously with facial surface shape and brain shape, the global-to-local segmentation approach used here yielded replicable associations that would otherwise have been missed by limiting to a global definition of the vault. While few additional hits were detected for the 14 segments that were not the global cranial vault, this could not have been known a priori given the poor understanding of normal-range cranial vault shape. Hence, we retained the segmentation approach to show that SNPs affecting vault shape mostly do so on a global level. Still, certain loci showed significant effects on

local segments only, whereby the association strength decreased gradually going from a local to a global level, though still with some degree of association with the global cranial vault. This is exactly what one should expect if the true effect of a locus is indeed local, i.e., that considering a larger set of vertices increases the total variance in the phenotype while the variance explained by the genotype does not increase with the same extent, thereby resulting in a statistically weaker association. One example is rs3936018 near *TBX15*, which was associated specifically with the frontal region, and previously identified to be associated with forehead morphology[32]. Another example is rs11609649 near *ALX1*, which also showed association locally with the frontal region, and not with the posterior regions, matching in situ expression patterns in mouse embryos[118]. While for those loci, we observed effects in the regions where they were expected, it is interesting to see the lack of any significant association in other regions of the vault, thereby affirming their localized roles.

Furthermore, we specifically examined the parietal versus frontal localization of candidate genes near our GWAS loci based on the pattern of their GWAS signal throughout the cranial vault segments, as well as RNAseq data from murine (E15.5) parietal and frontal bones. We found several of the GWAS signals in cranial segments to be prominently frontal or parietal while also showing expression differences in the same frontal and parietal mouse tissue. These trends add additional support that the human GWAS-based signals are functionally rooted to their role in cranium development in mouse embryos, affirming that some genes may have a predominant frontal or parietal contribution to cranial vault shape. A possible explanation may lie in the different cellular origins of both bones, the frontal being derived from the cranial neural crest and the parietal from the mesoderm. Nonetheless, *Fgf10*, *Zeb2*, and *Adamtsl3* showed a predominant parietal association with cranial vault shape while being downregulated in the parietal mouse tissue. It is possible that the parietal tissue is more sensitive to alterations in corresponding protein levels. If so, genetic variation relating to gene regulation would be expected to have a more pronounced effect despite the lower overall transcriptional activity relative to the frontal tissue. Functional assays could test this idea. Other explanations relate to how both methods record their signals, i.e., a GWAS captures accumulated effects over time, whereas RNAseq provides a snapshot in time. As such, RNAseq might fail to capture potentially more dominant effects of genes in different regions at different times. Furthermore, brain-related influences may substantially impact cranial vault shape and could not be picked up by the RNAseq.

Although most multi-ancestry genome-wide mapping efforts to date have been limited to GWAS meta-analysis[119–122] or admixture mapping[123–127], several works have intensively explored the idea of a joint multi-ancestry GWAS[37–39,128,129]. In general, those works have agreed that adjustment for global genomic ancestry is necessary to avoid spurious results and that adjustment for local genomic ancestry is useful for obtaining more accurate effect sizes and better signal localization in the presence of LD expansion that comes with admixture[130]. Because the ancestral make-up of the ABCD study cohort is highly heterogeneous and unbalanced (with a large group of European ancestry individuals), we opted to adjust for local genomic ancestry to ensure accurate results of downstream analyses and biological interpretations. Atkinson et al. (2021) have made recent efforts toward multi-ancestry GWAS. Their methodology, called *Tractor*, splits the allelic dosage of a SNP by the inferred donor ancestries and estimates different effect sizes for the different ancestral groups in a mixed ancestral cohort (including admixed). Its implementation can theoretically be extended to support any number of ancestral groups but is currently limited to univariate traits only. As with many statistical methods, multivariate implementations are likely to follow. However, accurate estimation of multivariate phenotypic coefficients (i.e., latent traits) for different ancestral groups based on a single multi-ancestry

cohort, is still likely to require many representatives from each ancestral group. This is therefore a limitation of the current study and similar studies to come.

We acknowledge that although multi-ancestry, our GWAS cohort is still predominantly comprised of individuals with recent European ancestry. While 4198 out of 6772 (62.0%) cohort subjects were assigned European ancestry, the prevalence of European ancestry in the admixed individuals contributed strongly to the overall local genomic European ancestry in the cohort. Hence, our discovery effort may be largely European-driven, and the improved discovery rate in our GWAS over a European-only GWAS may be due to the additional European-derived alleles. Nonetheless, we demonstrate that our GWAS hits are enriched for shared SNPs between the ancestral groups as a direct result of our multi-ancestry approach, and that effect sizes are mostly consistent across those groups, clearly illustrating the contribution of non-European inclusion. This also suggests that our GWAS successfully identified shared genetic components of cranial vault shape variation and implies that the phenotypic effects associated with our 30 lead SNPs likely hold more predictive value across populations[131]. With the high need for transferable results from GWAS and its derived applications, such as polygenic (risk) scores, running a single joint multi-ancestry and admixed GWAS as opposed to a stratified analysis may be a direct way to obtain more widely generalizable results.

## Methods

### Ethics statement

Use of patient data was approved by local ethics committees at U.C. Davis (IRBNet; protocol: 215635-25). Data available through controlled access repositories (UK Biobank, NIMH data archive) has been approved for broad sharing and local institutional approval (S63179, S60568, respectively) was granted for access to these datasets. All participants of these studies provided written informed consent to participate and have their data shared. Experiments resulting in RNA-seq data were approved by and performed under the oversight of the University of Pittsburgh Institutional Animal Care and Use Committee (protocol: 20057353).

### Discovery cohort

The Adolescent Brain Cognitive Development (ABCD) study[132] is a longitudinal study following brain development and health through adolescence. From data release 3.0 (February 2020), high-quality full-head MRIs were available for 11,702 participants. Following the ABCD recommendations for image inclusion, 471 individuals were removed from the dataset. Anthropometric data, including age, sex at birth, weight, and height were available on 11,787 individuals. Individuals with missing data ($n = 11$) or extreme outlier values (Z-score > 6; $n = 14$) for any of these variables were excluded. All participants were between 9 and 10 years old at the time of data collection and represent diverse ancestral backgrounds.

For this study, Quality Controlled Genotype Data from the ABCD cohort was downloaded as ABCD_release3.0_QCed Data files (.bed,.fam,.bim). It contained 11,099 unique individuals and 516,598 genetic variants aligned to the positive strand of build GRCh37. Subject information such as interview age, sex at birth, height, weight, BMI, and MRI information were extracted from the compiled RDS file (nda3.0.Rds) in the Data Exploration and Analysis Portal (DEAP).

### Genotyping and imputation of discovery cohort

The ABCD data files were first converted to a variant call file (VCF) using PLINK[133] 2.0. Reference SNP cluster IDs (rsIDs) were added using Bcftools[134,135] 'annotate' and dbSNP154.hg37. The ABCD data was then lifted and sorted from build GRCh37 to GRCh38 using Bcftools and in-house scripts. SNPs and indels were merged and only SNPs ($n = 502,882$) were kept for downstream phasing and imputation per

chromosome 1–22. Phasing was performed using default parameter settings from SHAPEIT v4.2.2[136] and the New York Genome Center (NYGC) 30x-1000 genomes-phased-dataset as reference, which can be found at [http://ftp.1000genomes.ebi.ac.uk/vol1/ftp/data_collections/1000G_2504_high_coverage/working/20201028_3202_phased/].

The reference data file used for imputation was prepared by the *Iliad* genomic data pipeline[137]. In short, the final reference dataset was compiled with data from the Human Genome Diversity Project (HGDP)[138] and the 1000 Genomes Project[139]. HGDP samples were retrieved as individual sequence alignment files in CRAM format [https://ftp.sra.ebi.ac.uk/1000g/ftp/data_collections/HGDP/data/]. The samples underwent variant calling via Bcftools 'mpileup' and 'call' commands for each chromosome. The HGDP dataset alone consists of 828 individuals from 54 populations. The data was reduced in variant size using an annotations list provided by the NYGC[140] which was input as a regions file. Finally, each HGDP combined chromosome VCF was phased and then merged with its chromosomal representative from the NYGC 30x-1000 genomes-phased-datasets to create a more globally represented reference set (NYGC1KGEN-HGDP) of 61,876,281 variants (including X chromosome) from 4030 Individuals in 80 populations that was used for imputation.

Imputation of the ABCD data was performed using IMPUTE5 v1.1.5[141] with SHAPEIT4 b38 genetic maps and the combined NYGC1KGEN-HGDP reference dataset using default parameter settings. Imputation regions that would maximize the use of both the reference and target sets were generated using imp5Chunker_v1.1.5[141]. Setting a threshold imputation INFO score of > 0.3, an ABCD imputed dataset of 11,099 individuals with $n = 49,745,078$ variants was generated. Following the ABCD recommendations, 82 individuals that were genotyped on plate 461 were excluded for further analysis.

## Genomic ancestry inference

Genotypes of unrelated ABCD participants (using King Robust[142] with a cutoff of 0.0442) were merged with 1000 G[139] Phase 3 and HGDP[138] genotypes of unrelated individuals. Markers in common were pruned iteratively, using PLINK[133] 2.0, using a window of 1000 markers, a step size of 50 markers and an $r^2$ cutoff of 0.2 until no more markers were being excluded. The resulting dataset was subjected to PCA to build an ancestry space, in which the relatives were then also projected. The first 10 PCs were used to express global genomic ancestry for each ABCD participant.

We defined local genomic ancestry to be discrete and assumed that each local genetic fragment can be traced back to a single ancestral population. In order to reduce model complexity and avoid potentially counterproductive errors, we limited local ancestry inference to three ancestral groups: European, African and American. With these groups, including any admixture thereof, we were able to include most of the ABCD participants. The following paragraphs describe how ABCD participants and ancestry references were selected for local ancestry inference.

In order to refine the combined set of 1000G and HGDP reference ancestry samples, we applied the ADMIXTURE[143] software in an unsupervised approach to estimate the genetic ancestry proportions of the individuals within the reference set for a given number of populations, *K*. The *K*:6 model was chosen as it best reflected continental distribution. It also allowed us to discern the samples from Oceania as one distinct ancestry component among the six (see cluster 5 in Supplementary Fig. 12).

Further analysis of the six co-ancestry proportions for this reference data included K-means clustering and sample filtering based on each designated cluster's main ancestry component to decrease the effect of noise with the clusters and create 'anchor' population reference data. K-means *elkan* algorithm from sklearn.cluster library on Python v3.7.6 designated each reference sample into one of six clusters based on their co-ancestry components. To develop these clusters

even further, a filter was applied to each cluster for retention of samples that displayed greater than the average proportion of the cluster's main ancestry component (e.g., if Cluster 0 had a K0 ancestry component average of 0.90 across all the Cluster 0 samples, only samples with a proportion of greater than 0.90 would be kept as a reference anchor). This effectively created the desired reference data of anchors to estimate the unknown biogeographical ancestries of the ABCD subjects. Analyses of the anchor reference data are shown in Supplementary Fig. 12 and sample sizes of the anchor reference data are provided in Supplementary Table 10. The European, African, and American anchor references were selected as ancestry representatives for local ancestry inference.

The supervised technique in ADMIXTURE[143] was used to estimate the co-ancestry proportions of the entire ABCD dataset from the anchor populations with a *K*:6 model. There were $n = 251,073$ variants after additional, independent QC (--hwe 1e-50 'midp' --maf 0.01) was performed on ABCD subjects prior to merging with the reference anchor samples. The co-ancestry proportions output from the supervised *K*:6 test was used to filter ABCD participants. Specifically, we retained 10,334 (10,334/11,099; 93.1%) participants with >90% joint European, African, and Indigenous American ancestry proportions for local ancestry inference. To identify European-ancestry participants, the co-ancestry proportions were subjected to K-means clustering, followed by additional filtering, similar to what was done to the anchor references as mentioned above, yielding 5746 participants with a nearly wholly European ancestry component.

Genome-wide local ancestry was then inferred for ancestry-prefiltered ABCD participants, using the 1000G and HGDP anchor references from the European (EUR; $n = 613$), African (AFR; $n = 545$), and Indigenous American (AMR; $n = 105$) clusters as ancestry representatives. This was done using RFMIX[144] v2 with one expectation-maximization step, a window size of 0.2 cM, and a terminal node size of 5. The expectation-maximization step reanalyzed admixture in the ancestry representatives to improve the ability to distinguish ancestries. A genomic map, to inform of switch positions was obtained from the Eagle[145] software website.

We observed good consistency between global ancestry estimated by ADMIXTURE[143] and RFMIX[144] (Pearson correlation coefficient > 0.998 for all ancestries). In addition, genome-wide local ancestry proportions doubled as a quality control for imputation, which to date is still a non-trivial task for multi-ancestry datasets. We reasoned that any systematic bias in haplotype matching during the imputation process would result in a detectable change in local genomic ancestry. Overall, genome-wide local ancestry proportions were consistent, except for some centromeric (e.g., chromosome 9) and telomeric (e.g., chromosome 21, 22) regions, as well as some known regions of long-range LD (e.g., chromosome 6), some of which have been previously reported to result in local ancestry biases[130,146–148] (Supplementary Fig. 13). In addition, no anomalies in local ancestry were detected at the 30 genome-wide significant GWAS loci (Supplementary Fig. 13).

## Image acquisition, phenotyping, and quality control

Minimally processed T1w structural MRI data were downloaded from the ABCD data repository. Image preprocessing steps include distortion correction, movement correction, resampling (1 mm isotropic voxels), alignment to standard space, and initial quality control[149]. Facial surfaces can be segmented from MR images using intensity thresholding-based approaches. However, resulting segmentations are prone to errors due to noise, partial volume effect, placeholders, and MR bias fields. More specifically, the ABCD MRI data contains artifacts, such as missing parts and noise, that are unstructured over the different subjects and acquisitions. Hence, these artifacts can be reduced prior to segmentation, given a sufficient number of independent samples. Instead of re-acquiring MRI data for the same 'target' subject and averaging them out to remove the artifacts, we generated virtual

re-acquisitions by an inter-subject intra-MRI non-rigid image-based registration approach. A total of 300 MRI scans ('floating' scans)—matched in terms of sex at birth, height, weight, and genomic ancestry—were registered to a single target MRI scan using Elastix (SimpleITK library[150] in Python) with the Param0000 parameter map (affine and B-spline)[151]. The use of 300 floating scans per 'target' image was chosen based on visual inspection of the results while controlling for computational time and resources. The resulting, denoised consensus 'target' image was defined as the voxel-per-voxel median of the resulting warped 'floating' images.

Based on the work of Matthews et al. (2018)[152] a full head template ($n = 28,218$ vertices) was constructed as the average of the expected head shapes of boys and girls at 9.5 years old, i.e., those closest in age to our study cohort. The cranial vault ($n = 11,410$ vertices) was manually delineated on this template, encompassing the supraorbital ridge and extending towards the occipital bone (Supplementary Fig. 1).

Denoised MR scans were imported as.nii files into Matlab (version 2021a) and the craniofacial surface was extracted as a mesh using the isosurface function (Matlab function: *isosurface*). Full head surfaces were cleaned by cropping internal head structures, also captured by the volumetric imaging protocol, based on the distance of each vertex to the centroid. The full head template was then registered to the extracted craniofacial surfaces using MeshMonk[153] following a scaled rigid and non-rigid registration step. Prior to rigid registration, an initial rough alignment was performed to ensure a similar orientation of both template and target scans. Because all MR scans were already aligned to standard space during image preprocessing, this could be done by manually placing five positioning landmarks on 100 randomly selected target surfaces and transferring the averages of these landmarks to all individual scans.

Visual inspection of the data showed that compression of soft tissue structures caused by external fixation systems during the MRI scanning procedure was primarily observed near the ears, cheeks, and chin. To remove such distortions, ears were flagged as outliers during Meshmonk surface registration. In addition, while focusing on the face and cranial vault (i.e., the most reliable parts of the face), erroneous regions were interpolated using thin-plate splines in Matlab.

For all participants, the cranial vault shape was extracted from the full head meshes based on the selected region in Supplementary Fig. 1. Images were manually inspected and labeled according to the type of error/artifact observed (e.g., partially missing data due to cropping of the MRI scans, external structures such as goggles, and remaining distortions of the head surface). The set of good-quality images was then further inspected for outliers due to mapping errors or still overlooked imaging artefacts. Similar to previous works[29,32], we measured the Mahalanobis distance for each cranial vault to the overall average cranial vault in a shape space spanned by PCs that capture 98% of total shape variation. Based on the distribution of Mahalanobis distances, a z-score was calculated for each image, and only images with an absolute z-score lower than 1.5 were considered for further analysis, yielding a set of 9015 well-QCed images.

### Global-to-local segmentation of cranial vault shape

To study genetic effects on cranial vault variation at both a global and local level, we performed a data-driven hierarchical segmentation on symmetrized cranial vault shape, similar to earlier work on facial and brain shape[29,32,33]. First, the meshes were adjusted for age, sex at birth, height, weight, cranial size, the scanner (encoded by dummy variables), and the 10 first genomic PCs.

Next, a vertex-wise similarity matrix was constructed by measuring the correlation of the distance-to-centroid between each pair of vertices. This measure was augmented with pairwise Euclidian distance (proximity in 3D) to make nearby vertices more similar to enforce connectivity and coherence within the segments. Specifically, both measures were normalized separately so that the most alike pair of vertices (highest correlation or smallest distance) got a value of 1, and the least alike pair got a value of 0. These were then combined using a weighted sum of 60% correlational similarity and 40% proximity-based similarity. These percentages were chosen based on connectivity and coherence of segments resulting from different weighted sums of both measures at 5% intervals. Hierarchical spectral clustering[29] was carried out based on the resulting similarity matrix. At each step the vertices were split into two maximally similar clusters, resulting in 1, 2, 4, and 8 non-overlapping segments at hierarchical levels 1, 2, 3, and 4 respectively. For each of the resulting segments, independently of the other segments, the configurations were aligned using generalized Procrustes analysis (GPA) and subjected to PCA. Parallel analysis[154,155] was used to determine the number of PCs required to explain the major phenotypic variance with fewer variables.

### Final discovery GWAS cohort

The intersect of ABCD participants which satisfied covariate quality control, genotype quality control ($n = 11,017$), ancestral pre-filtering ($n = 10,334$), and image quality control ($n = 9015$) yielded 8217 participants. From this set, we removed 1445 relatives using the King Robust[142] software with a cutoff of 0.0442 (3rd degree relatives), resulting in a final GWAS sample size of 6772 (3742 male and 3030 female) of which 4198 (2220 male and 1978 female) were assigned European ancestry and 2504 non-European, predominantly mixed ancestry. Imputed variants of ancestry-selected participants ($n = 10,334$) were filtered for SNPs only, with MAF > 1% and missing genotyping rate > 95%, resulting in a final set of 10,647,531 SNPs.

### Joint multi-ancestry and admixed GWAS approach

At each SNP, cranial vault shape across all 15 segments was adjusted for the assigned local ancestral origin of its respective genomic region, coded additively for two out of three ancestries (European, African, and Indigenous American ancestry). This adjustment for local genomic ancestry aimed to remove phenotypic differences due to ancestry-associated genotyped or ungenotyped variation, such that any phenotypic effect measured is dose-dependent on the SNP, independent of genomic ancestry (globally and locally). In other words, we wanted to ensure that we found genotype-phenotype associations at marker-level resolution to ensure accuracy of downstream analyses.

Under the reduced model, the residual phenotypic variance in each of 15 cranial vault segments was tested for association with the presence of the major allele, coded based on the additive genetic model. This was done using CCA, which extracts the linear combination of PCs maximally correlated with the SNP, representing a multidimensional shape deformation in the PC space. Significance was tested using a right-tailed chi square test with degrees of freedom equal to the number of phenotypic PCs (Supplementary Fig. 14).

Genome-wide significance of genotype-phenotype associations was declared at the traditional threshold of $P < 5e{-}8$. However, because of the increased multiple-testing burden introduced by multiple GWAS runs on different cranial vault segments, we empirically estimated the effective number of independent tests based on genotype permutations, following the procedure of Kanai (2016)[156]. Briefly, for a single SNP we randomly permuted the genotypes 10,000 times, essentially creating the distribution of chance associations with cranial vault shape under the null hypothesis. Each permuted genotype was tested for association with the 15 cranial vault segments and the lowest $P$-value was retained. We then divided 0.05 by the 5th percentile of the resulting 10,000 $P$-values to estimate the number of effective GWAS runs performed. This was repeated for 500 random SNPs, resulting in an average effective number of phenotypes of 11.44 (SD: 0.56). The effective number of phenotypes was additionally estimated following the same approach using the 30 lead SNP as well as using three eigenvalue-based methods[157–159] on the 15 × 15 segment-segment correlation matrix as estimated using either genome-wide $P$-values, chi

squared statistics, or genomic Pearson correlations, yielding 10 additional estimates in the range of 8.52–11.28 (See Supplementary Note for details on additional approaches for estimating the effective number of phenotypes). We then opted for the former, most conservative estimate, and obtained a subsequent 'study-wide' significance threshold of $P < 4.37e-9$ (i.e., 5e−8/11.44).

No sex stratified GWAS was performed in this work for reasons related to power, however, we did adjust for sex assigned at birth and resulting GWAS thus focused on aspects of cranial vault shape variation independent of sex.

## Peak detection and annotation

Peak detection was performed in three steps starting from 1658 SNPs that reached genome-wide significance in the GWASs. Starting from the most significant SNP, all SNPs within 250 kb, as well as those within 1 Mb and in LD ($r^2 > 0.01$) were clumped into a single locus represented by the most significant SNP (lead SNP). This was repeated until all SNPs were assigned a locus ($n = 32$). Next, any two loci were merged into a single locus if the lead SNPs were within 10 Mb and in LD with $r^2 > 0.01$. The lead SNP with the lowest $P$-value was chosen to represent the newly merged locus. Lastly, for robustness any singleton SNP below the study-wide significance threshold was removed ($n = 0$). This resulted in 30 genome-wide and 24 study-wide significant genomic loci associated with cranial vault shape. LD was calculated using the genotypes of the GWAS cohort. LocusZoom[40] plots for all 30 loci are available in the Supplementary Data 1.

To study functional enrichments of the genes near our 30 GWAS loci, we performed Gene Ontology (GO) enrichment analysis using GREAT[42] v4.0.4 with default settings. Significance was determined at 5% FDR by both binomial and hypergeometric tests. To look for enrichments of phenotypes in GWAS Catalog, we used FUMA[160] v1.3.7 with default settings and the list of genes obtained from GREAT for consistency across the enrichment analyses.

Next, a set of candidate genes at each locus was compiled by combining the genes annotated by GREAT v4.0.4[42], FUMA[160] v1.3.7, as well as manually annotated genes within 500 kb of the lead SNP based on available literature (Supplementary Tables 8 and 9). Specifically, we relied on evidence of involvement in craniofacial development from mouse models and human craniofacial syndromes, as well as previous implications in GWAS on the cranial vault or face (the forehead in particular). Lastly, we performed Bayesian colocalization analysis[161] (R package 'coloc') to check if our GWAS hits coincided with an expression quantitative trait locus (eQTL) in any of 22 tissues selected from GTEx8 based on their relevance to the cranial vault (Supplementary Table 11). Prior probability distributions were set by default. Any gene within 1 Mb of the lead SNP was considered a candidate gene at the locus if the posterior probability of locus overlap ('PP4') was ≥70%. Additionally, all eQTLs with PP4 ≥ 50% are listed in Supplementary Table 12.

## Calculating genomic inflation

Following the original definition of the genomic inflation factor[162], we calculated an overall inflation factor for our combined set of 15 GWASs (one corresponding to each cranial vault segment) as the median observed test statistic divided by the median expected test statistic under the null hypothesis (i.e., no genotype-phenotype association). As SNPs in the combined GWAS were represented by their lowest $P$-value across the 15 segments, a null distribution of $P$-values was obtained empirically by repeating the GWAS with randomly permuted genotypes for each SNP and taking its lowest $P$-value across the segments. Because $P$-values corresponding to each segment were derived from chi square distributions with different degrees of freedom, we obtained a normalized test statistic, $\nu$, form the chi square statistics following Naqvi et al. (2021)[33] as $\nu = \chi_D^2 / D (1 + \frac{\chi_D^2}{N})$, with $D$, the degrees of

freedom, and $N$, the sample size. The genomic inflation factor was then calculated as the median observed test statistic divided by the median test statistic from the empirical null distribution.

## Deformation hotspots

For each of the 21 SNPs with genome-wide significant ($P < 5e-8$) effects on global cranial vault shape, vertex-wise normal distances were calculated between the average cranial vault and the cranial vault transformed along the latent shape direction associated with the SNP. The absolute values were then normalized to unit length per SNP and a vertex-wise distribution of deformation magnitudes was subsequently obtained. At each vertex, the 95th percentile of the SNP-associated deformation magnitude was used for visualization in Supplementary Fig. 3.

## Replication of cranial vault shape variants

The UK Biobank[35] contains genetic and phenotypic data on approximately 500,000 UK volunteers, aged 40–70 at recruitment. Full-head T1-weighted MR images were available for 39,609 subjects from release v2.3 (October 2020). These scans were obtained in anonymized form, meaning that the entire face and regions around the ears were removed. The process of anonymization has been described in detail by Alfaro-Almagro et al.[163].

The MRI data in NifTi format was imported into Matlab (version 2021a), and the isosurface was extracted (function: *isosurface* with an *iso* value of 250) from each image, followed by taking the concave hull (function: *boundary* with shrink factor set to 1) to remove internal structures. A cranial vault template was aligned to each mesh based on four landmarks located at the most superior, posterior, and lateral (left/right) points of the head. We then performed rigid and non-rigid surface registration, using MeshMonk[153] while mitigating the damage from the MRIs. Specifically, we masked all vertices in the damaged regions (Supplementary Fig. 5) across all images and restricted these vertices from contributing to the deformation of the template during mapping, essentially treated the damaged regions as missing data. Recompletion of the missing regions was done by statistical data imputation, where an active shape model[164] constructed from the processed ABCD vault data was used to impute the position of the masked vertices. To further mitigate any impact of the damage on the surface registration, non-rigid deformations of the template were constrained by the active shape model, i.e., it replaced the more general freeform non-rigid deformation model in the MeshMonk[153] toolbox. This resulted in cranial vault shapes that realistically represented cranial vaults according to the active shape model but lacked individual level variation in the damaged regions. Consequently, the segment-wise replication rate was decreased based on the extend of data treated as missing (Supplementary Fig. 5).

The quality of the mapping relied on the rigid alignment of target and template, which in turn was hindered by the damage done to the images. In some cases, this caused the template to rotate incorrectly. During image quality control, rotations measured along all three orthogonal directions were combined into a single chi square statistic with 3 degrees of freedom. This statistic was then normalized by taking the right-tailed cumulative density so that unrotated images were assigned 1, asymptotically decreasing to 0 for heavily rotated images. Using a cut-off value of 0.8, 17,959 (unrotated) images were retained (Supplementary Fig. 15).

All subjects with missing covariates or covariate outliers (>6 SD) were removed. Ancestral outliers (>3 SD) were removed based on the 4 first genomic PCs ('elbow' of Scree plot). The remaining cranial vaults ($n = 17,214$) were adjusted for age, age squared, weight, height, sex, cranial size, scanner, and the first 10 genomic PCs using PLSR. The residuals were added to the average ABCD cranial vault shape, followed by applying the same global-to-local segmentation as defined in the ABCD cohort. Segments were then independently aligned using

GPA and projected into the corresponding ABCD shape space per segment. Subjects without genotype data, or with more than 5% missingness were omitted from further analysis, as well as relatives up to the third degree identified using the King[142] robust algorithm with a cutoff of 0.0442. This resulted in a sample size of 16,846 (7439 male and 9407 female) for the final replication cohort. Out of the 30 genome-wide significant discovery lead SNPs, 29 overlapped with the replication dataset directly; rs34898775 was selected as a proxy SNP for rs202055590 (located 1.1 kb away, with $r^2$ of 0.997 in all combined 1000 G populations).

To measure the presence of the associated shape trait from the discovery panel (ABCD), the replication panel (UK Biobank) was projected onto the latent shape trait, identified by CCA, for a particular SNP in a particular cranial vault segment. The resulting univariate scores were calculated for each lead SNP/segment pair ($n = 108$) for which significant ($P < 5e{-}8$) associations were found. The F-statistic from a linear regression was calculated to determine the significance of association between the projection scores and the SNP genotypes for all 108 SNP-segment combinations. A 5% FDR-adjusted $P$-value threshold of $P < 0.0244$ was calculated using the Benjamini-Hochberg[165] method, yielding 55/108 significant tests, and 20/30 significant SNP replications in at least one cranial vault segment.

## $F_{ST}$ enrichment analysis

The Weir and Cockerham estimators[166] for Wright's fixation index, $F_{ST}$ were calculated per SNP between European (EUR) and African (AFR), and EUR and Indigenous American (AMR) samples in the 1000 G Phase 3 dataset using vcftools[167] v0.1.17. Next, $F_{ST}$ enrichment analysis[36] was performed on our set of GWAS lead SNPs to see if the GWAS signals were significantly enriched for high or low $F_{ST}$ SNPs. The average $F_{ST}$ across the lead SNPs was compared to a distribution of 10,000 averaged $F_{ST}$ values, each calculated based on the same number of SNPs with matched MAF and LD scores[43]. To match the LD scores to the GWAS cohort, covariate-adjusted LD scores were calculated based on the 6772 GWAS cohort subjects with a window size of 20 cM (--ld-wind-cm 20) and after adjusting genotypes for the first 10 genomic PCs using scripts of Luo et al. (2021)[168]. For matching SNPs, the range of MAFs and LD scores was divided into 20 equally spaced quantiles, and a matched SNP was selected at random from the same bins as the original SNP. Significance of the enrichment was declared if the observed average $F_{ST}$ was lower than the 2.5th percentile or higher than the 97.5$^{th}$ percentile of the null distribution (5% alpha).

## Calculating heterogeneity of effect size

For each of the 30 lead SNPs, we performed $2 \times 2 \times 2$ tests of effect size heterogeneity between ancestry groups, with each test imposing different restrictions on a generative model. First, we considered univariate versus multivariate SNP-trait associations. Under the univariate ($\bar{y}, \beta_{1-5} \in \mathbb{R}$) scenario, the latent associated shape trait from the main GWAS was tested (Eq. 1), while under the multivariate ($\bar{\mathbf{y}}, \beta_{1-5} \in \mathbb{R}^{1 \times m}$, with $m$ the number of phenotypic dimensions) scenario, the latent trait was free to vary (Eq. 2). Second, we considered both 2-way and 3-way heterogeneity of effect size. Under the 2-way scenario, only African versus European effect size heterogeneity ($\beta_3, \beta_4 \in \mathbb{R}$ vs. $\beta_3 = \beta_4 \in \mathbb{R}$, with $\beta_2 = \beta_5 = 0$) was tested, and under the 3-way scenario, African versus European versus Indigenous American effect size heterogeneity ($\beta_3, \beta_4, \beta_5 \in \mathbb{R}$ vs. $\beta_3 = \beta_4 = \beta_5 \in \mathbb{R}$) was tested. Third, we looked for heterogeneous effects sizes in the most significantly associated cranial vault segment, as well as all significant ($P < 5e{-}8$) segments. Under the *best segment* scenario, heterogeneity of effect size was only tested in the segment most significantly associated with the SNP in the GWAS, while under the *significant segments* scenario, heterogeneity of effect size was tested in all segments where that SNP was significant ($P < 5e{-}8$) in the GWAS, and the lowest 5% FDR-adjusted $P$-value was kept.

$$\bar{y} = \beta_0 + \beta_1 a_{EUR} + \beta_2 a_{AFR} + \beta_3 x_{EUR} + \beta_4 x_{AFR} + \beta_5 x_{AMR} \qquad (1)$$

$$\bar{\mathbf{y}} = \boldsymbol{\beta_0} + \boldsymbol{\beta_1} a_{EUR} + \boldsymbol{\beta_2} a_{AFR} + \boldsymbol{\beta_3} x_{EUR} + \boldsymbol{\beta_4} x_{AFR} + \boldsymbol{\beta_5} x_{AMR} \qquad (2)$$

To test for statistical significance, we considered a full model (with parameters $\theta$), which included different effect sizes for different ancestries, and a constrained model (with parameters $\theta_0$) with equal contributions from all ancestries. Significance of $-2{*}\ln\left(\frac{\mathscr{L}_{\theta_0}}{\mathscr{L}_\theta}\right)$ was determined based on the upper tail of a $\chi^2$ distribution with degrees of freedom equal to the difference in number of parameters estimated between the nested models. Observe that the constraint model represents the standard GWAS setting, albeit with additional adjustment for local ancestry, where a biallelic SNP is modeled under the additive genetic model, i.e., its state is modeled as 0, 1, or 2, counting the major allele. The full model is equivalent to *tractor* by Atkinson et al. (2021)[129], where alleles are assigned a donor ancestry, and are then modeled under the additive genetic model per ancestry. Tractor scripts were used to extract ancestry-specific allelic dosages, $x_i$. Local ancestry is modeled by $a_i$. The univariate phenotype, $\bar{y}$ and the multivariate phenotype, $\bar{\mathbf{y}}$ were pre-adjusted for all covariates, including global genomic ancestry using PLSR.

## GWAS in a single-ancestry versus multi-ancestry cohort

To allow for comparative analysis between the multi-ancestry GWAS and a single-ancestry GWAS, a European-only GWAS was conducted on a subset of 4198 samples with assigned European ancestry, representing the largest ancestral group within the total cohort. This GWAS was run analogously to the full, multi-ancestry GWAS, albeit without adjusting phenotypes for local genetic ancestry. Peaks were then called analogously to the full, multi-ancestry GWAS, and loci were considered to be overlapping between both GWASs if corresponding lead SNPs were within 250 kb. Shape effects and segment-wise association profiles were then summarized for each pair of overlapping loci. When no lead SNP was within 250 kb in the other GWAS, the locus was considered to be specific to one GWAS. In that case, we elected the SNP within 250 kb that had the lowest $P$-value in the GWAS to represent the locus.

## Localization of genetic effects using RNA-seq in E15.5 mice

Timed-pregnant female CD1 mice were purchased from Charles River Laboratory. The mice were housed under standard conditions in the University of Pittsburgh Division of Laboratory Services vivarium. The E15.5 embryos were collected via C-section after $CO_2$ euthanasia of the pregnant dam. Sex was not considered when selecting the embryos. All animal husbandry, procedures and protocols were approved by and performed under the oversight of the University of Pittsburgh Institutional Animal Care and Use Committee. We dissected 9 E15.5 skull vaults along the coronal suture into the frontal (frontal bones and metopic suture), and the parietal (parietal bones and sagittal suture) components. Each biological replicate consists of the frontal and parietal components from 3 embryos therefore no sex-stratified analysis was possible. The tissue was lysed using with Trizol (Invitrogen), followed by additional homogenization using the Qiashredder and then the RNA was extracted with the Qiagen RNAeasy kit. Axeq made and sequenced the libraries from 6 biological replicates ($n = 3$ frontal, and $n = 3$ parietal) as 100 bp paired end reads on an Illumina HiSeq2000 sequencer (Axeq Technologies, South Korea).

The resulting 6 FASTQ formatted paired-read sequence files produced by Axeq Technologies were subsequently downloaded and visually inspected for quality control purposes with FASTQC[169] v0.11.9. Read trimming was performed by Trimmomatic v0.32[170] using a sliding window of four nucleotides and a mean Phred score ($Q$ value) greater

than 20 (SLIDINGWINDOW:4:20) corresponding to an error probability of 0.01 over the four nucleotides. Sequences shorter than 25 nucleotides (MINLEN:25) were removed.

The resulting trimmed reads were then aligned to the Ensembl primary assembly mouse reference Genome, GRCm39, available at [http://ftp.ensembl.org/pub/release-106/fasta/mus_musculus/dna/], utilizing the GRCm39 gene annotation file, available at: [http://ftp.ensembl.org/pub/release-106/gtf/mus_musculus/] using the STAR[171] sequence aligner version 2.7.10a. Alignment for the 6 trimmed paired-end reads showed an average uniquely mapped read percentage of 94.3% ± 0.2%.

Gene expression quantification was carried out using the GenomicAlignment R package[172] set with the following options: *Mode = Union*, *singleEnd = FALSE*, *ignore.strand = TRUE*, *fragments = FALSE*. Normalization of differentially expressed gene counts were based on five stable housekeeping genes (*Actb*, *Gapdh*, *Rer1*, *Rpl27*, and *Rpl13a*) identified in the Ho and Patrizi, 2021 mouse cranium and brain development study[173]. To improve visualization of gene expression, the normalized gene counts were variance stabilizing transformed (VST)[174–176], which is roughly similar to transforming the data to the $\log_2$ scale, allowing for better visualizations. To allow for easier comparisons between genes, especially for the combined expression plots, the mean difference in expression, hereafter referred to as 'mean difference', of the sample grouping to all other (parietal and frontal) sample groupings were calculated and plotted. This allows the plots to report the $-\log_2$ expression change of each tissue-specific group (e.g., Sample Group 1 is sampled from the frontal region and thus labelled as 'frontal') to the mean of all other samples (e.g., the mean of Sample Groups 1–6). The R package DESeq2 1.36[174,177] was used to analyze the expression data using the DESeq function[174,177], which transforms read counts based on the estimation of size factors and dispersion, fits to a negative binomial GLM, performs a Wald significance test, and assesses based on a cutoff false discovery rate of <0.05 using the Benjamini-Hochberg procedure[165]. All bioinformatic analyses were performed using R-4.2.1 and Bioconductor[178].

As part of the validation of GWAS hits via differential analysis, we compared the phenotypes utilized in the GWAS to those of the two tissues sampled in the expression analysis. Since the expression analysis was based on tissues taken from the parietal and frontal areas of the mouse embryo while the GWAS utilized 15 cranial vault segments, to directly compare the phenotypes we classified the 15 GWAS segments as containing either primarily 'frontal' or primarily 'parietal' content. As some segments, like segment 1, contain both parietal and frontal content, those segments were labelled as 'both.' Subsequently, this allowed candidate genes near the GWAS lead SNPs to be classified as either frontal, parietal, or a combination of both parietal and frontal, depending on the cranial vault segment in which the lead SNPs were initially found. One issue of this classification scheme is that lead SNPs were not always identified in the most phenotypically specific cranial vault segment (i.e., hierarchical levels 3 or 4). To remedy this, we performed several classifications. One classification scheme, labelled as 'most significant hit', based the classification of candidate genes on the most significant segment in which it was found (i.e., based on the classification of segments found in hierarchical levels 1–4). The other classification, labelled as 'most specific hit', utilized the more phenotypically specific segments in which the candidate gene was either found in or directly related to. In this classification, a label with equal or higher specificity was obtained recursively by considering the most significantly associated 'child' segment until the encountered segment comprised solely of frontal or parietal content or until hierarchical level four was reached. As an example, GWAS lead SNP rs17479393 was identified in segments 1, 2, 3, 7, 8, and 15. The 'most significant hit' classification simply takes the most significant segment, segment 1

in this example, and assigns a classification. As segment 1 encompasses both the parietal and frontal regions, it is labelled as 'both'. The 'most specific hit' classification starts at segment 1 and follows the strongest association of its 'child' segments to reach segment 8 (through segments 2 and 4), which contains solely parietal bone content.

Frontal and parietal tissue gene expression from mice embryos were analyzed to generate expression profiles that were ultimately compared to human candidate genes located near lead SNPs identified in the GWAS analysis. To make this comparison possible, gene symbol and human homolog mappings were carried out on mouse Ensembl identification IDs using the Biomart R package[179].

### Enrichment for enhancer activity
Signals of acetylation of histone H3 on lysine K27 (H3K27ac) in the vicinity of the 30 genome-wide significant lead SNPs were calculated as described in White & Indencleef et al. (2021)[32]. Information regarding the cell types and tissues used together with a link to where they are available can be found in Supplementary Table 5. Briefly, to compare H3K27ac signal in the vicinity of the genome-wide significant lead SNPs between cell-types in an unbiased manner, we divided the genome into 20 kb windows, and calculated H3K27ac reads per million (RPM) from each aligned read (bam or tagAlign) file in each window using bedtools coverage (v2.27.1). We then performed quantile normalization (using the normalize.quantiles function from the preprocessCore package, v3.7) on the matrix of 154,613 windows × 133 ChIP-seq datasets. We then selected the windows containing each of the 30 genome-wide significant lead SNPs.

### Genetic overlap with brain and facial traits
To quantify sharing of genetic signals between a pair of GWAS, we calculated Spearman correlations as described in Naqvi et al. (2021)[33]. Briefly, SNPs were first selected to overlap with the HapMap3 SNPs[180], and SNPs within the major histocompatibility complex were removed. We then organized the remaining SNPs in 1725 LD blocks that are approximately independent in individuals of European ancestry[181]. The mean SNP $-\log_{10}(P\text{-value})$ was calculated per block, and a rank-based Spearman correlation was calculated on the averaged association score per block. A standard error of the correlation coefficient was estimated based on 100 bootstrapping cycles.

### Testing variants for their role in NCS
We tested for genotype-phenotype associations between our 30 genome-wide significant SNPs and craniosynostosis based on a dataset comprising whole genome sequence data (WGS; dbGaP, phs001806.v1.p1) from families with children affected with four different types of craniosynostosis – coronal (72 trios; $n = 276$), lambdoidal (17 trios; $n = 51$), metopic (62 trios; $n = 186$), and sagittal (63 trios; $n = 189$). These families are trios with sequence data on an affected child and two unaffected parents. Trios were analyzed using the transmission disequilibrium test (TDT) via PLINK[133]. Trios were only analyzed with trios of the same phenotype (i.e., metopic craniosynostosis families were only analyzed with other metopic families) so we had four discrete analyses based on the four types of craniosynostosis. We then checked the results of the 30 genome-wide significant cranial vault SNPs in these four analyses.

### Reporting summary
Further information on research design is available in the Nature Portfolio Reporting Summary linked to this article.

## Data availability
Full cranial vault GWAS summary statistics for this study have been deposited to the NHGRI-EBI GWAS Catalog [https://www.ebi.ac.uk/

gwas/] under accession codes GCST90270327–GCST90270341 (one accession number per segment). All the data and detailed information for the ABCD Study, including MRI scans, genetic markers, and covariates are available under restricted access through the ABCD data repository [https://nda.nih.gov/abcd/] upon completion of the relevant data use agreements. The ABCD data repository grows and changes over time. The ABCD data used in this report came from data release 3.0 [https://doi.org/10.15154/1519007 and https://doi.org/10.15154/1528459]. All the data and detailed information for the UK Biobank data set, including MRI scans, genetic markers, and covariates are available under restricted access to bona fide researchers. Access can be requested via the UK Biobank data access process [https://www.ukbiobank.ac.uk/enable-your-research/apply-for-access]. The NYGC 30×1000 genomes phased dataset and HGDP dataset are freely available online [http://ftp.1000genomes.ebi.ac.uk/vol1/ftp/data_collections/1000G_2504_high_coverage/working/20201028_3202_phased/, and https://ftp.sra.ebi.ac.uk/1000g/ftp/data_collections/HGDP/data/]. The WGS data from the craniosynostosis cohorts used in this study is available from dbGaP under accession code phs001806.v1.p1. The mouse cranial bone RNAseq dataset used in this study is available in the GEO database under accession code GSE245664. The mouse GRCm39 reference genome assembly and gene annotation file used in this study are available from Ensembl [http://ftp.ensembl.org/pub/release-106/fasta/mus_musculus/dna/ and http://ftp.ensembl.org/pub/release-106/gtf/mus_musculus/]. The cis-eQTL data from 22 tissues used in this study are available from the GTEx V8 database [https://gtexportal.org/home/datasets]. The LD block coordinates used in this study are available from Berisa et al. at [https://bitbucket.org/nygcresearch/ldetect-data/src/master/]. The H3K27ac ChIP-seq datasets used in this study are available from the Gene Expression Omnibus and Roadmap Epigenomics databases. Accession codes and links can be found in Supplementary Table 5. Source data for the manuscript figures, 3D animations of cranial vault effects, and the anthropometrics masks used in this study are available from our FigShare repository [https://doi.org/10.6084/m9.figshare.c.6858271.v1][34]. Source data are provided with this paper.

## Code availability

KU Leuven provides the MeshMonk v.0.0.6 spatially dense facial-mapping software, free to use for academic purposes available at [https://github.com/TheWebMonks/meshmonk] and from our FigShare repository [https://doi.org/10.6084/m9.figshare.c.6858271.v1][34]. Matlab implementations of the hierarchical spectral clustering to obtain facial segmentations are available from a previous publication [https://doi.org/10.6084/m9.figshare.7649024.v1][182].

The statistical analyses in this work were based on functions in Matlab 2021a, python v3.7.6, R v4.2.1, PLINK 2.0, bcftools v1.10.2, vcftools v0.1.17, SHAPEIT v4.2.2, IMPUTE5 v1.1.5, imp5Chunker v1.1.5, ADMIXTURE v1.3.0, RFMIX v2, MeshMonk v0.0.6, GREAT v4.0.4, FUMA v1.3.7, LocusZoom, FASTQC v0.11.9, Trimmomatic v0.32, STAR sequence aligner v 2.7.10a, Bioconductor, bedtools v2.27.1, R libraries (GenomicAlignment, DESeq2 v1.36, Biomart, preprocessCore v3.7, coloc v5.1.0.1, locuscomparer v1.0.0, circlize v0.4.15), python packages (SimpleITK v 2.1.0), scripts from Luo et al. (2021, available at: https://github.com/immunogenomics/cov-ldsc), and scripts from Atkinson et al. (2021; available at: https://github.com/Atkinson-Lab/Tractor), as mentioned throughout the Methods.

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

## Acknowledgements

Data used in the preparation of this article were obtained from the Adolescent Brain Cognitive Development (ABCD) Study (https://abcdstudy.org), held in the NIMH Data Archive (NDA). This is a multisite, longitudinal study designed to recruit more than 10,000 children age 9–10 and follow them over 10 years into early adulthood. The ABCD Study is supported by the National Institutes of Health and additional federal partners under award numbers U01DA041048, U01DA050989, U01DA051016, U01DA041022, U01DA051018, U01DA051037, U01DA050987, U01DA041174, U01DA041106, U01DA041117, U01DA041028, U01DA041134, U01DA050988, U01DA051039, U01DA041156, U01DA041025, U01DA041120, U01DA051038, U01DA041148, U01DA041093, U01DA041089, U24DA041123, U24DA041147. A full list of supporters is available at https://abcdstudy.org/federal-partners.html. A listing of participating sites and a complete listing of the study investigators can be found at https://abcdstudy.org/consortium_members/. ABCD consortium investigators designed and implemented the study and/or provided data but did not necessarily participate in analysis or writing of this report. This manuscript reflects the views of the authors and may not reflect the opinions or views of the NIH or ABCD consortium investigators. This research has been conducted using the UK Biobank Resource under application number: 43193. The resources and services used in this work were provided by the VSC (Flemish Supercomputer Center), funded by the Research Foundation – Flanders (FWO) and the Flemish Government. This work was funded in part by the Intramural Research Program of the National Human Genome Research Institute, National Institutes of Health (A.M.M., C.M.J.). This project was supported in part by the National Institute of Dental and Craniofacial Research: R01DE027023 (S.M.W., J.R.S., P.C., J.W.); R01DE016886 (S.A.B., P.A.R.); R03DE031061 (S.A.B.); K99DE032729 (S.N.). Support for whole genome sequencing of the craniosynostosis cohort provided through X01HL14053 (S.A.B.).

## Author contributions

P.C., S.M.W., M.D.S., J.R.S., J.W., S.W. and S.G. conceptualized the study. S.G., H.H., M.Y., P.C., S.W., N.H., R.J.E., S.N., A.M.M., C.M.J., S.A.B. and H.L.S-R. carried out the data curation. S.G., H.H., R.J.E., S.N., D.V. and P.C. wrote software. S.G., H.H., R.J.E, A.M.M, S.W., S.N. and N.H. carried out the formal analysis. S.G., H.H., R.J.E., S.W., A.M.M, S.N., N.H., S.M.W., P.C., H.L.S-R. and M.Y. carried out the investigation. S.G., H.H., S.N., R.J.E., M.K.L., N.H, and P.C. did the visualization. P.C., S.M.W, S.W., J.W., M.D.S., J.R.S., M.L.M., P.A.R. and S.A.B. were responsible for funding acquisition. P.C., S.M.W, S.W., J.W., M.D.S. and J.R.S. carried out the supervision. S.G., S.M.W., H.H., R.J.E., S.W., S.N., A.M.M., H.L.S-R., N.H. and P.C. wrote the original draft. S.G., H.H., R.J.E., A.M.M., C.M.J., M.Y., S.N., N.H., M.K.L., D.V., H.L.S-R., P.A.R., S.A.B., M.L.M., J.R.S., M.D.S., J.W., S.W., S.M.W. and P.C. reviewed and edited the final manuscript.

## Competing interests

The authors declare no competing interests.
