## [Peer Review File · Nature Communications]

Joint multi-ancestry and admixed GWAS reveals the complex genetics behind human cranial vault shapeREVIEWER COMMENTS

Reviewer #1 (Remarks to the Author):

In this paper, Goovaerts et al. performed multivariate genome-wide associations on the morphologies of cranial vault, using an ancestrally heterogeneous imaging study (Adolescent Brain Cognitive Development Study, ABCD) as the discovery cohort. After controlling for global ancestries and local ancestries of the ABCD samples, they found 30 independent loci that reached genome-wide significant threshold. The authors performed series of follow-up analyses, including replications in a single ancestry cohort (UKBiobank), association tests on case trio data, sensitivity analyses on the effect sizes (Fst and heterogeneity test), GO enrichment analyses, annotation overlap analyses, and genetic overlaps with face and brain measurements. In general, this paper is well written and the analyses are comprehensive. It highlights the complex dynamics for morphological features originated from the ectoderms, affirming important roles of several cranio-facial related genes. I only have some minor comments.

1. The authors have performed series of sensitivity analyses on the effects of including global ancestry and local ancestry in the model, as well as using stratified approach. Indeed, as the author pointed out, using the entire heterogeneous cohort while controlling for both global and local ancestry detect the shared signals across populations and thus avoid potential confounds originated from the gene by environment correlations. However, it is unclear if the improved power is driven just by increasing sample size (6K vs 4K) rather than what the author claimed by inclusion of multi-ancestry groups. As the authors showcased, the lack of heterogeneous effects and differential allele frequencies indicate their discovered loci is shared and homogeneous across populations. It means the multi-ancestry does not provide additional ancestral specific effects here.
2. The loci overlapping across the global-local features of face, brain, and cranial vaults are fascinating. The results and the discussion on this complex interaction are a bit dense and hard to unpack, despite the importance and richness of their materials. I would recommend updating the infographics of their Figure 5 to show which genes are shared across regions and which are more local specific, and what are their conjectured directions.
3. Unlike the authors' papers about GWAS on face and GWAS on brain shape, this paper did not examine the loci overlaps with other polygenic disorders. Although current emphases on linking the discovered genes with the known cranial abnormalities are more important than the polygenic pleiotropies with other complex traits, it would be good to know the rationale behind this decision from the authors.
4. The Table 2 used "the highest activity" to describe the where the strongest signals are observed and the "biased activity" to describe the differential expressions. Not sure if that is the best term to use in the context.

Reviewer #2 (Remarks to the Author):

The authors conducted a multi-ancestral GWAS together with a European replication study on human outer head surface and identified 30 genetic loci of which they replicated 20. While this reviewer appreciates the authors' efforts in unveiling the genetic basis of human head shape variation, there are several issues that make it difficult to know how novel and how reliable the reported findings are.

- 1) As phenotype basis for their GWAS, the authors used the outer surface of the head obtained from MRI, but report all their GWAS results for the cranial vault. They defend their approach by stating that the outer head surface represents a proxy for the cranial vault, but do not present evidence for this. What is the empirical evidence for this assumption? Why do they claim their results in connection with a proxy phenotype and not the phenotype they measured? This must be

elaborated so that the reader can put the reported genetic findings in the right phenotypic perspective.

2) Performing multi-ancestry GWAS is much less straightforward than single-ancestry GWAS and comes with several issues that can lead to false positive findings. This reviewer wonders how meaningful the multi-ancestry GWAS is in this case, where the majority of the discovery samples and all the replication samples are of single ancestry i.e., European and only a relatively small proportion of the discovery samples is of non-European ancestry. How do these limited non-European samples impact on the genetic findings? It is important to see the full results based on European-only samples and compare them with those from the multi-ancestry samples to get a practical idea how the intrinsic problems of multi-ancestry GWAS impacted on the presented results. The authors very briefly mention a comparison between multi-ancestry discovery and European-only discovery, but it seems this analysis was focussed on outcomes of the multi-ancestry GWAS and in any case was not presented with the needed level of detail. Both analyses, European-only and combined, should be reported and the differences critically discussed.

3) The segmentation approach the authors applied, which they developed previously regarding the face, results in several segments of the outer head surface for which the biological relevance appears largely questionable if not being absent at all. This reviewer cannot see that the segmentation approach was done with a biological hypothesis in mind, which in principle should not be done when aiming to study a biological phenomenon. Notably, most of the GWAS findings they reported were obtained for "global vault shape", i.e., the total outer head surface as measured without any segmentation. Only very few hits were seen for any of the 14 segments generated from the total surface with the segmentation approach. This reviewer largely questions the use of the segmentation approach for this phenotype and remains very sceptical about the true positive nature of the genetic loci reported solely for segments not overlapping with those identified with total shape. Moreover, doing GWAS on multiple phenotypes comes for the price of having to consider multiple phenotypes and their correlation in the analysis and interpretation, which the authors mostly ignored (see next point). Perhaps a GWAS on the single phenotype of total outer head surface would be more powerful and deliver more trustworthy results?

4) The authors performed 15 separate GWASs for the 15 phenotypes they used from applying their segmentation approach. It seems they did not do a meta-analysis on the outcomes of these 15 GWASs, which would be the normal way to do. Since meta-analysis is a well-established approach, the question appears why the authors deviated from this? If the authors were to stick with the multiple phenotypes they generated with their segmentation approach despite the criticism expressed under point 3, meta-analysis should be conducted and significant study results should only be concluded from the meta-analysis outcomes. Also, it can be assumed that there are correlations between these 15 phenotypes used, which, in case the authors keep using them, should be tested empirically and such correlations, if identified, should be taken into account in the GWAS in an appropriate statistical way.

5) As far as this reviewer can see, the authors left out describing which of the genetic loci they identified were discovered for the first time for their phenotype(s), therefore representing new knowledge, and which were already identified in previous studies and represent confirmation of previously established knowledge. It is unexpected that the authors did not look into this given the novelty focus of scientific journals. Why did they not describe this? Given that this is not the first genetic study on head shape, it should be clearly described how many and which loci represent novel knowledge and how many and which confirm previous knowledge.

Reviewer #3 (Remarks to the Author):

The manuscript by Goovaerts et al. describes a large-scale GWAS analysis of the human cranial vault shape in the ABCD study cohort with replication in individuals from the UK biobank. The manuscript is very comprehensive in scope and while many of the identified loci and candidate genes have previously been implicated in craniofacial development, the findings significantly add to our understanding of the genetic factors that influence the cranial vault shape.

Major comments:

- Hierarchical spectral clustering was used to segment the cranial vault into 15 distinct segments.

It would be useful to understand to what extent the segments correspond to (or overlap) cranial vault bones and whether edges of the segments coincide with cranial sutures.

- The extent of damage in the validation dataset and its potential impact on the replication analysis of each segment is unclear and should be better explained. For example, an overview of the proportion of repaired MRI profiles per segment should be provided. It is also not clear to what extent the repair recapitulates the expected shape without validation (e.g. by creating damaged profiles and then checking to what extent repair recapitulates the expected shape). Depending on the impact of the repair, may be better to exclude segments with too much damage altogether.
- The differential gene expression analysis shows only minor changes between the frontal and parietal bones, with log₂ fold changes ranging between -1.37 and 0.68. While these differences may be statistically significant it is not sufficient to classify genes as 'frontal' or 'parietal'. Statements such as "This consistency between gene expression and phenotypic effect could suggest that for those genes increased expression drives phenotypic effects." (line 375) and "This may indicate that the downregulation of FGF10 allows for an increased activity of other genes involved in shaping the cranial vault" (Line 383) should therefore be moderated.
- Candidate gene assignment relied on GREAT and FUMA analyses combined with manual curation. GTEX could provide further supporting evidence for a link to gene expression in e.g. skeletal tissues.

Minor comments:

- Line 165, change 'genome-wide significant' to 'genome-wide significance'.
- It would be useful to expand supplementary table 1 to include the coordinates of the 30 genome-wide significant loci identified after merging.
- Supplementary table 4 is uninterpretable due to the large size. It should be provided as a spreadsheet, perhaps with an additional tab that limits the results to only the significant expression differences.
- "Other candidate genes were either not differentially expressed, had fewer than one gene count, or were not expressed at E15.5.". Does 'fewer than one gene count' refer to the entire library or e.g. fewer than one count-per-million reads? Also, what is the threshold used to differentiate between expressed and non-expressed genes?

REVIEWER COMMENTS

Reviewer #1 (Remarks to the Author):

In this paper, Goovaerts et al. performed multivariate genome-wide associations on the morphologies of cranial vault, using an ancestrally heterogeneous imaging study (Adolescent Brain Cognitive Development Study, ABCD) as the discovery cohort. After controlling for global ancestries and local ancestries of the ABCD samples, they found 30 independent loci that reached genome-wide significant threshold. The authors performed series of follow-up analyses, including replications in a single ancestry cohort (UKBiobank), association tests on case trio data, sensitivity analyses on the effect sizes (Fst and heterogeneity test), GO enrichment analyses, annotation overlap analyses, and genetic overlaps with face and brain measurements. In general, this paper is well written and the analyses are comprehensive. It highlights the complex dynamics for morphological features originated from the ectoderms, affirming important roles of several cranio-facial related genes. I only have some minor comments.

1. The authors have performed series of sensitivity analyses on the effects of including global ancestry and local ancestry in the model, as well as using stratified approach. Indeed, as the author pointed out, using the entire heterogeneous cohort while controlling for both global and local ancestry detect the shared signals across populations and thus avoid potential confounds originated from the gene by environment correlations. However, it is unclear if the improved power is driven just by increasing sample size (6K vs 4K) rather than what the author claimed by inclusion of multi-ancestry groups. As the authors showcased, the lack of heterogeneous effects and differential allele frequencies indicate their discovered loci is shared and homogeneous across populations. It means the multi-ancestry does not provide additional ancestral specific effects here.

Author response:

With the power-deficit of differentiated SNPs, it is likely that a European-only GWAS with the same sample size yields a greater discovery rate over the multi-ancestry one. Therefore, the main aim of this sensitivity analysis was to see if it was worthwhile conducting a multi-ancestry GWAS over a European-only GWAS. We are unsure if the increased discovery rate is due to boosted sample size alone or any other aspects related to multi-ancestry inclusion. Consequently, we carefully reworded parts of the text to avoid suggesting that non-European ancestry was responsible for the improved discovery rate, but rather that it was something we observed in one GWAS versus the other: *“Overall, the multi-ancestry GWAS thus achieved a higher discovery rate, identifying 7 more loci, and reaching lower P-values in 25 of the 31 loci (80.6%).”*

Indeed, most loci show no significant heterogeneity of effect between ancestral groups. For those that do, we could not accurately estimate the shape effects in the minority ancestries due to the limited sample size. Admixture mapping is a technique that specifically looks for phenotypic differences associated with genetic differentiation in an admixed cohort and could theoretically identify genetic loci underlying ancestry differences in cranial vault shape. We initially tried admixture mapping in the ABCD cohort during early stages of this work but ran into statistical problems related to the multivariate phenotypic definition of cranial vault shape as well as the limited admixed sample size after filtering and the complex admixture patterns in the cohort. Therefore, at the current time, with the current data and models, admixture mapping did not contribute any added value to the manuscript, however, we plan on revisiting the approach in future work.

2. The loci overlapping across the global-local features of face, brain, and cranial vaults are fascinating. The results and the discussion on this complex interaction are a bit dense and hard to

unpack, despite the importance and richness of their materials. I would recommend updating the infographics of their Figure 5 to show which genes are shared across regions and which are more local specific, and what are their conjectured directions.

Author response:

We made changes to Figure 5 to make it more comprehensive and easier to unpack. First, we moved panels b (vault-wide genetic correlations with facial segments) and c (vault-wide genetic correlations with brain segments) to a supplementary figure. We also added a chord diagram showing all 22 loci overlapping with the face (White et al. 2021) or brain (Naqvi et al. 2021). Therein we show which genes are shared between which segments between the different structures.

3. Unlike the authors' papers about GWAS on face and GWAS on brain shape, this paper did not examine the loci overlaps with other polygenic disorders. Although current emphases on linking the discovered genes with the known cranial abnormalities are more important than the polygenic pleiotropies with other complex traits, it would be good to know the rationale behind this decision from the authors.

Author response:

Earlier work indeed demonstrated genetic overlap between several neuropsychiatric/cognitive traits and cortical shape, but not facial shape. While their relationship with the brain is more direct, their relationship with the face, if any, is much weaker. By similarity, we expect any genetic overlap between neuropsychiatric/cognitive traits and normal-range cranial vault shape to be small. Either way, we determined that this was outside the scope of the current study.

Nonetheless, we included an Extended Data Table with enriched phenotypes from GWAS Catalog for the set of genes annotated to our GWAS loci to provide an unbiased overview of traits with an overlapping genetic basis. Specifically, these additional results highlight the overlap with other polygenic skeletal phenotypes and cranial vault shape, which we included in the manuscript as:

"In line with the associated mouse phenotypes, we found enriched biological processes related to bone development (skeletal system development, osteoblast differentiation, bone development, ossification, osteoblast development) and cartilage development (regulation of cartilage development, chondrocyte differentiation, cartilage development, positive regulation of cartilage development). Furthermore, the analysis yielded enrichments for mesenchyme-related terms (mesenchymal cell differentiation, mesenchyme development, connective tissue development), as well as more broad terms related to embryonic development (embryo development, embryonic morphogenesis, embryonic organ morphogenesis and development, ...). In agreement with these terms, enrichment of phenotypes in GWAS Catalog using FUMA yielded bone-related traits, including, bone mineral density (heel, skull, lumbar spine, ...) and height (Supplementary Table 4)."

4. The Table 2 used "the highest activity" to describe the where the strongest signals are observed and the "biased activity" to describe the differential expressions. Not sure if that is the best term to use in the context.

Author response:

We made appropriate changes to the wording. Table 1 now uses 'RNAseq label' and 'GWAS label'. We further avoided 'highest activity' and 'biased activity' in the text by using 'stronger effect' when talking about GWAS effects and 'higher transcriptional activity' when talking about differential expression. Phrases that were changed, include:

From: *the GWAS analysis identified a higher activity in the bone ...*

To: *the GWAS analysis identified a stronger effect in the region ...*

From: *which all showed a frontal biased activity ...*

To: *which all showed higher transcriptional activity in the frontal bone ...*

Reviewer #2 (Remarks to the Author):

The authors conducted a multi-ancestral GWAS together with a European replication study on human outer head surface and identified 30 genetic loci of which they replicated 20. While this reviewer appreciates the authors' efforts in unveiling the genetic basis of human head shape variation, there are several issues that make it difficult to know how novel and how reliable the reported findings are.

1) As phenotype basis for their GWAS, the authors used the outer surface of the head obtained from MRI, but report all their GWAS results for the cranial vault. They defend their approach by stating that the outer head surface represents a proxy for the cranial vault, but do not present evidence for this. What is the empirical evidence for this assumption? Why do they claim their results in connection with a proxy phenotype and not the phenotype they measured? This must be elaborated so that the reader can put the reported genetic findings in the right phenotypic perspective.

Author response:

Craniofacial malformations such as craniosynostosis demonstrate how variations in skull shape are clearly visible through the outer head surface. This should be no surprise since the soft tissue thickness surrounding the neurocranium is thin and does not vary much in thickness as illustrated in Figure 3 (<https://doi.org/10.1371/journal.pone.0210257.g003>) from Gietzen et al.¹. Therefore, it is possible to measure shape variation associated with the neurocranial bones through the outer head surface. This is reflected in our findings, which clearly demonstrate the skeletal roles of genes at our identified loci.

Fig 3. From Gietzen et al. 2019 – A method for automatic forensic facial reconstruction based on dense statistics of soft tissue thickness – Statistic of the FSTT on a mean skull. Mean and standard deviation of FSTT computed from the 43 CT scans.

(<https://doi.org/10.1371/journal.pone.0210257.g003>)

Furthermore, in practice, measures of the boney cranial vault are routinely taken on the soft-tissue envelope. For example, in traditional anthropometry, measures of cranial vault shape are obtained by applying calipers to soft-tissue landmarks². In fact, these landmarks (glabella, opisthocranium, eurion) have the same names on the skull and immediately overlying soft tissue due to their close correspondence.

Because the vault refers to a broad anatomical region which may include the skull alone or the skull plus its soft tissue envelope, a precise phenotypic definition is required and given to avoid confusing the reader. We had previously already done so in the first sentence of the results section and in Extended Data Figure 1: “Cranial vault shape, herein defined as the outer head surface encompassing the supraorbital ridge and extending towards the occipital bone, was extracted from structural MRIs (Extended Data Fig 1; Methods)”

To further clarify that a regional definition for the cranial vault is used, we also modified the first sentence of the introduction to read:

“The cranial vault – the globular portion of the head, shaped by flat, plate-like bones that surround and protect the brain (...)”

Furthermore, we avoid using the term ‘proxy’ altogether, and changed:

“Due to their close correspondence, the outer soft-tissue vault surface served as a proxy for the underlying cranial bones.”

To:

“Since the outer soft-tissue layer in this region is thin and uniform, shape variation associated with the neurocranial bones is well captured by our phenotypic definition.”

2) Performing multi-ancestry GWAS is much less straightforward than single-ancestry GWAS and comes with several issues that can lead to false positive findings. This reviewer wonders how meaningful the multi-ancestry GWAS is in this case, where the majority of the discovery samples and all the replication samples are of single ancestry i.e., European and only a relatively small proportion of the discovery samples is of non-European ancestry. How do these limited non-European samples impact on the genetic findings? It is important to see the full results based on European-only samples and compare them with those from the multi-ancestry samples to get a practical idea how the intrinsic problems of multi-ancestry GWAS impacted on the presented results. The authors very briefly mention a comparison between multi-ancestry discovery and European-only discovery, but it seems this analysis was focussed on outcomes of the multi-ancestry GWAS and in any case was not presented with the needed level of detail. Both analyses, European-only and combined, should be reported and the differences critically discussed.

Author response:

The ABCD cohort is an ancestrally diverse cohort with only 5,746 out of 11,099 (51.8%) European participants based on the ancestry inference described in the manuscript. The other 48.2% of participants were mostly admixed, had very little Asian ancestry, and could not be further stratified into any substantial parental populations. Therefore, the aim of this work has always been to conduct a multi-ancestry study as opposed to omitting almost half of the cohort in a Euro-centric study. While the literature is rapidly expanding with regards to multi-ancestry meta-analysis in stratified cohorts, literature on joint multi-ancestry and admixed GWAS remains relatively scarce. Thus, as a GWAS in an ancestrally complex US sample our study represents a valuable contribution to the existing literature.

Upon initial submission, our manuscript described how the inclusion of 2,504 non-European samples in a cohort of 6,772 US participants

- yielded stronger GWAS signals at the majority of loci shared with a European-only GWAS;
- yielded 8 additional genome-wide significant GWAS signals, while losing one relative to the European-only GWAS;
- reduced statistical power for high F_{st} SNPs;
- resulted in an enrichment of shared SNPs across ancestries;
- and yielded consistent effect sizes across ancestries.

We have now expanded the multi-ancestry versus European GWAS comparison. In a supplementary document we show that for each locus the latent shapes and association profile (i.e., the pattern of P-values across hierarchical segments) is highly concordant between both GWASs. Additionally, we show colocalization of the signal at each locus using LocusZoom and LocusCompare plots.

Furthermore, we updated the infographic that compared signal strength in both GWASs to be less focused on the multi-ancestry GWAS and better show which loci were shared or not. In conclusion,

we have extensively demonstrated how the inclusion of non-European individuals affected genetic findings. This paragraph was extensively reworked in the text and now reads:

“We next performed a sensitivity analysis to evaluate the discovery of genome-wide loci when including or excluding the 2,504 non-European-ancestry GWAS cohort subjects. To this end, a locus was considered to be overlapping between both GWASs if the corresponding lead SNPs were within 250 kb, resulting in a joint set of 31 unique loci (Extended Data Fig 7). Among these loci, 22 were shared, of which 17 (17/22; 77.3%) reached lower P-values in the multi-ancestry GWAS (Extended Data Fig 7). Moreover, 8 loci were unique to the multi-ancestry GWAS, and one locus (rs563186113) was unique to the European-only GWAS. Nonetheless, all those 9 loci still showed some degree of association in the other GWAS. Overall, the multi-ancestry GWAS thus achieved a higher discovery rate, identifying 7 more loci, and reaching lower P-values in 25 of the 31 loci (80.6%). Furthermore, shape effects and segment-wise association profiles were highly concordant between both GWASs at each of the 31 loci (Supplementary Information 2). At 21 loci (21/31; 67.7%), the same segment generated the lowest P-value, and at 5 more loci (5/31; 16.1%) the lowest P-values were generated in segments that were directly related. Strong co-localization was observed for the genomic signals in both GWASs as shown by LocusZoom and LocusCompare plots in Supplementary Information 2. Altogether, the analyses conducted in this section extensively demonstrated which aspects of the genetic findings were influenced by the inclusion of non-European individuals”

3) The segmentation approach the authors applied, which they developed previously regarding the face, results in several segments of the outer head surface for which the biological relevance appears largely questionable if not being absent at all. This reviewer cannot see that the segmentation approach was done with a biological hypothesis in mind, which in principle should not be done when aiming to study a biological phenomenon. Notably, most of the GWAS findings they reported were obtained for “global vault shape”, i.e., the total outer head surface as measured without any segmentation. Only very few hits were seen for any of the 14 segments generated from the total surface with the segmentation approach. This reviewer largely questions the use of the segmentation approach for this phenotype and remains very sceptical about the true positive nature of the genetic loci reported solely for segments not overlapping with those identified with total shape. Moreover, doing GWAS on multiple phenotypes comes for the price of having to consider multiple phenotypes and their correlation in the analysis and interpretation, which the authors mostly ignored (see next point). Perhaps a GWAS on the single phenotype of total outer head surface would be more powerful and deliver more trustworthy results?

Author response:

Since the aim of the study was to study cranial vault shape variation, we reasoned that shape variation could directly inform the segmentation, resulting in an unbiased and data-driven approach, such as used in earlier work on the face and brain. Additionally, the segmentation approach works in conjunction with PCA, i.e., grouping correlated vertices yields more efficient dimensionality reduction, aiding with association testing. This approach represents only one of many ways to segment the cranial vault. The resulting segment edges do not coincide with cranial sutures, however, that does not indicate a lack of biological meaning. Fundamentally, a data-driven segmentation will capture biological aspects of shape variation that are not immediately obvious to a single human observer.

To attempt to expose the underlying biological drivers of the segmentation, we examined the vertex-wise shape effects of all lead SNPs with genome-wide significant effects on global cranial vault shape, i.e., those SNPs whose discovery was not dependent on the segmentation approach. Specifically, we looked at the regions on the cranial vault where those SNPs effects were the largest by looking at the 95th percentile of absolute shape deformation associated with each vertex. This revealed deformation hotspots, many of which located on the midline, and thus clearly coinciding

with sutures. Interestingly, overlaying our data-driven segmentation revealed how the deformation hotspots get more isolated and focused on within each hierarchical layer of segmentation. Strikingly, the smallest segments were centered around specific hotspots. Hence, we show how our data-driven segmentation has plausible biological underpinnings and is clearly influenced by suture-associated shape variation. Extended Data Figure 3 shows these segments and hotspots.

As we have shown previously with facial surface shape and brain shape, the global-to-local segmentation approach used here yields replicable associations what would otherwise have been missed by limiting to a global definition of the vault. While it is true that few additional hits were seen for the other 14 segments (all segments, excluding the full cranial vault), this could not be known a priori given the poor understanding of normal-range cranial vault shape. Hence, we retained the segmentation approach to show that SNPs affecting vault shape mostly do so on a global level, which is a result in itself. Still, certain loci showed strong effects on local segments only. Examples include rs3936018, near *TBX15* associated specifically with the frontal region ($P = 4.10e-19$), and previously identified to be associated with forehead morphology³. Another example is rs11609649 near *ALX1*, which also showed association locally with the frontal region ($P = 8.24e-10$), and not with the posterior regions, matching *in situ* expression patterns in mouse embryos (<https://www.informatics.jax.org/assay/MGI:3508780>). While for those loci, we observed effects in the regions where they were expected, it is interesting to see the lack of significant association in other regions of the vault and affirming their localized roles. Moreover, we identified a signal near *SHH*, which was only locally associated with cranial vault shape, has a well-established role in craniofacial development, and was replicated in the UKBB.

Regarding the remark that loci with local effects still show some association for the global cranial vault, Supplementary Information 1 clearly shows that the global shape effects of those loci are consistent with the local shape effects. Also note that the association profile across the segments is not random, often showing a gradient of association strength from local to global segments. This is exactly what one should expect if the true effect of a locus is indeed local, i.e., that considering a larger set of vertices increases the total variance in the phenotype while the variance explained by the genotype does not increase with the same extent, thereby resulting in a statistically weaker association. We have updated the infographics in Supplementary Information 1 to better show this. Additionally, we indicated which segments were replicated in the UKBB.

4) The authors performed 15 separate GWASs for the 15 phenotypes they used from applying their segmentation approach. It seems they did not do a meta-analysis on the outcomes of these 15 GWASs, which would be the normal way to do. Since meta-analysis is a well-established approach, the question appears why the authors deviated from this? If the authors were to stick with the multiple phenotypes they generated with their segmentation approach despite the criticism expressed under point 3, meta-analysis should be conducted and significant study results should only be concluded from the meta-analysis outcomes. Also, it can be assumed that there are correlations between these 15 phenotypes used, which, in case the authors keep using them, should be tested empirically and such correlations, if identified, should be taken into account in the GWAS in an appropriate statistical way.

Author response:

Many GWAS meta-analyses assume that there is no sample overlap between GWASs, such as Fisher's P, Stouffers Z, and inverse-variance weighted (IVW) meta-analysis (which additionally assumes the same effect size across traits). This assumption is violated in our study, as samples fully overlap between each of the 15 GWASs. Furthermore, most other approaches require (signed) effect size estimated and their standard errors (e.g., METAL, MTAG, metaCCA, C-GWAS) which are not yielded by our multivariate regression using CCA. Hence, while meta-analysis of GWAS is common

practice, most of the current implementations are simply not compatible with studies like ours where there is complete sample overlap and no signed effect size estimates (and their standard errors) are available. We therefore went with an approach that makes no assumptions on sample overlap, requires only P-values, and has been applied before in similar studies (Naqvi et al., White et al.). An overview of meta-analysis methods is given in the table below.

Method	Citation	Assumptions	Input	Why this method is not applicable to our cranial vault GWAS
Fisher's P / Stouffers Z		No sample overlap	P-values	Samples overlap fully between segments.
Inverse variance weighted / Fixed effect meta-analysis		No sample overlap, beta constant across traits	Signed Z-scores (or signed betas), standard errors on beta	Samples overlap fully between segments. CCA does not yield z-scores, nor betas. Effects likely differ across segments.
minP	Tippett (1931)	Samples can overlap	P-values, individual level data (to calculate N_{eff})	Applicable!
METAL	Willer (2010) ⁴	Samples can overlap	Signed Z-scores or beta + SE (based on IVW)	CCA does not yield signed betas
Empirical Brown's Method / Kost's Method	Poole (2016) ⁵	Samples can overlap	P-values, individual level univariate traits (can be adapted)	Applicable!
metaCCA	Cichonska (2016) ⁶	Samples can overlap	Beta + SE	CCA does not yield betas
MTAG	Turley (2018) ⁷	Samples can overlap (Relies on bivariate LD-score regression to estimate sample overlap)	Signed Z-scores or beta + SE (based on IVW)	Relies on LD-score regression: - No standardized implementation for multivariate test statistics - Does not work in multi-ancestry cohort.
Cauchy test	Liu (2020) ⁸	Samples can overlap	P-values	Applicable!
C-GWAS	Xiong (2022) ⁹	Samples can overlap	P-values, signed betas (based on IVW)	CCA does not yield signed betas

To combine our GWAS results across multiple segments, we take for each SNP the minimal P-value across the 15 segments (minP). Since correlation between the segments is expected (due to e.g., the hierarchical nature of segments), a Bonferroni correction based on 15 independent tests would be overly stringent. For a more accurate adjustment, we estimated the number of effective independent tests per SNP following Kanai (2016) and ran 10,000 genotype-phenotype associations under the null by permutation testing in a way that preserved the correlational structure of our phenotypes. This was repeated for 500 random SNPs and yielding an estimate of 11.44 (± 0.55) independent traits.

To illustrate the robustness of this estimate, we have additionally estimated the number of effective traits using several other approaches:

First, we repeated the permutation testing for our 30 lead SNPs, yielding an estimate for the number of effective traits of 11.28 (± 0.50).

Second, we applied several methods to estimate the number of effective traits from a trait-correlation matrix. We used three methods: 1) Li & Ji (2005)¹⁰, 2) Galwey (2009)¹¹, and 3) Li (2011)¹² on three different correlation matrices: 1) based on the P-values from the GWAS, 2) based on the Chi-squared statistics from the GWAS, and 3) based on the Spearman genetic correlation matrix (methods). All three correlation matrices yielded highly concordant results across methods.

Method	P-values	Chi-squares	Genetic correlation
Li & Ji (2005)	11.00	10.00	10.00
Galwey (2009)	11.27	10.84	10.98
Li (2011)	8.87	8.52	8.57

While these estimates mostly agree on a number of effective traits, the most conservative estimate of 11.44 was obtained based on 10,000 permutations of 500 random SNPs. We therefore adjusted the genome-wide significance threshold (instead of the P-values themselves) using Bonferroni correction based on 11.44 effective traits, resulting in a study-wide threshold of $4.37e-9$ (i.e., $5e-8 / 11.44$).

In short, we tried several approaches to estimate the number of effective traits, thereby taking into account the correlational structure between the phenotypes. We showed robustness of our estimates and used the most conservative estimate to adjust the significance threshold.

5) As far as this reviewer can see, the authors left out describing which of the genetic loci they identified were discovered for the first time for their phenotype(s), therefore representing new knowledge, and which were already identified in previous studies and represent confirmation of previously established knowledge. It is unexpected that the authors did not look into this given the novelty focus of scientific journals. Why did they not describe this? Given that this is not the first genetic study on head shape, it should be clearly described how many and which loci represent novel knowledge and how many and which confirm previous knowledge.

Author response:

Only *HMGA2* was previously identified in two separate GWASs on head circumference. None of the other loci were previously identified in GWAS on cranial vault dimensions (head circumference, maximum cranial length, maximum cranial length, and cephalic index). This has now been indicated in the text:

Among these signals, only the locus near HMGA2 has been previously identified in GWAS on cranial vault dimensions.

As already mentioned in the text, several loci overlap with the facial GWAS of White et al. 2021. These loci and their associated segments are shown in the updated Figure 5.

Reviewer #3 (Remarks to the Author):

The manuscript by Goovaerts et al. describes a large-scale GWAS analysis of the human cranial vault shape in the ABCD study cohort with replication in individuals from the UK biobank. The manuscript

is very comprehensive in scope and while many of the identified loci and candidate genes have previously been implicated in craniofacial development, the findings significantly add to our understanding of the genetic factors that influence the cranial vault shape.

Major comments:

- Hierarchical spectral clustering was used to segment the cranial vault into 15 distinct segments. It would be useful to understand to what extent the segments correspond to (or overlap) cranial vault bones and whether edges of the segments coincide with cranial sutures.

Author response:

Generally, the segment edges of our data-driven approach do not coincide with sutures.

To help understand how cranial vault shape variation influenced the hierarchical segmentation, we examined the vertex-wise shape effects of all lead SNPs with genome-wide significant effects on global cranial vault shape, i.e., those SNPs whose discovery was not dependent on the segmentation approach. Specifically, we looked at the regions on the cranial vault where those SNPs effects were the largest by looking at the 95th percentile of absolute shape deformation associated with each vertex. This revealed deformation hotspots, many of which located on the midline, and thus clearly coinciding with sutures. Interestingly, overlaying our data-driven segmentation revealed how the deformation hotspots get more isolated within each hierarchical layer of segmentation. Strikingly, the smallest segments were centered around specific hotspots. Extended Data Figure 3 shows these segments and hotspots.

Therefore, it appears that the data-driven segmentation captures units of shape variation as these local foci of deformation rather than individual bones.

- The extent of damage in the validation dataset and its potential impact on the replication analysis of each segment is unclear and should be better explained. For example, an overview of the proportion of repaired MRI profiles per segment should be provided. It is also not clear to what extent the repair recapitulates the expected shape without validation (e.g. by creating damaged profiles and then checking to what extent repair recapitulates the expected shape). Depending on the impact of the repair, may be better to exclude segments with too much damage altogether.

Author response:

First, we reference the original work that describes in detail the defacing of the MRIs so that the reader can easily understand the extent of the damage. We added the following sentence to the first paragraph of the methods:

"The process of anonymization has been described in detail by Alfaro-Almagro et al.."

Because all MRIs were defaced using the same procedure, i.e., by masking voxels of the face (including forehead) and ears, the corresponding local vault segments (e.g., 5, 6, 7, 11, 12, 13) and all bigger vault segments (e.g., 1,2,3) were at least partially damaged in every single scan. Therefore, any approach that relies on the use of intact, undamaged segments would not be productive, which is also the reason that the UKBB was used as a replication dataset, not as a discovery dataset.

While almost all segments were partially damaged in all MRIs, they were still rich in individual level shape variation, especially those segments where the damage affected only a small fraction of the total surface area (e.g., segment 1). This is evidenced by the fact that, despite this damage, we were able to replicate 20 of 30 GWAS signals. We therefore approached this problem as a missing data problem, explicitly modeled the damaged regions as missing, and filled in the missing data based on the most plausible shape variation as modelled by an active shape model built from the ABCD vault

data. In other words, we filled in the damaged regions by an average, expected shape instead of trying to recapitulate the shape that was originally present. We appropriately changed wording to make this clearer in the text:

“Specifically, we masked all vertices in the damaged regions (Extended Data Fig 4) across all images and restricted these vertices from contributing to the deformation of the template during mapping, essentially treated the damaged regions as missing data. Recompletion of the missing regions was done by statistical data imputation, where an active shape model¹⁵⁹ constructed from the processed ABCD vault data was used to impute the position of the masked vertices. To further mitigate any impact of the damage on the surface registration, non-rigid deformations of the template were constrained by the active shape model, i.e., it replaced the more general freeform non-rigid deformation model in the MeshMonk¹⁵⁰ toolbox. This resulted in cranial vault shapes that realistically represented cranial vaults according to the active shape model but lacked individual level variation in the damaged regions. Consequently, the segment-wise replication rate decreased based on the extend of data treated as missing (Extended Data Fig 4).”

Additionally, we included panel b in Extended Data Figure 4 to show the fraction of vertices in each segment that was masked and treated as missing. Moreover, panel d clearly illustrates how the replication rate decreases when the fraction of missing data within a segment increases. A replication rate of 0 is expected when an entire segment is treated as missing, thus showing no inflation of the replication rate.

In conclusion, it was not our aim to recomplete damaged parts as close to the original shapes as possible. Instead the procedure we used deals with the damaged parts as missing data, and the results illustrate as expected, that a higher rate of missing data in a segment, leads to a lower replication rate. This is now illustrated clearly in the revised document.

- The differential gene expression analysis shows only minor changes between the frontal and parietal bones, with log2 fold changes ranging between -1.37 and 0.68. While these differences may be statistically significant it is not sufficient to classify genes as ‘frontal’ or ‘parietal’. Statements such as “This consistency between gene expression and phenotypic effect could suggest that for those genes increased expression drives phenotypic effects.” (line 375) and “This may indicate that the downregulation of FGF10 allows for an increased activity of other genes involved in shaping the cranial vault” (Line 383) should therefore be moderated.

Author response:

The sentence:

“This consistency between gene expression and phenotypic effect could suggest that for those genes increased expression drives phenotypic effects.”

has now been removed altogether.

The sentence:

“This may indicate that the downregulation of FGF10 allows for an increased activity of other genes involved in shaping the cranial vault.”

has been replaced by:

“The strong parietal localization of the FGF10 GWAS signal combined with a lower transcriptional activity may indicate differential FGF10 dosage sensitivity between both tissues, where the developing parietal bone could be more sensitive to small changes in FGF10 levels. Alternative explanations exist and exact mechanisms could be a subject for functional follow-up studies. Altogether, our data suggests that some genes may have a predominant frontal or parietal contribution to cranial vault shape.”

Additionally, in the discussion:

“Nonetheless, *Fgf10*, *Zeb2*, and *Adamts13* showed a predominant parietal association with cranial vault shape while being downregulated in the parietal mouse tissue. This may indicate that their down-regulation drives phenotypic effects.”

was changed to:

“Nonetheless, *Fgf10*, *Zeb2*, and *Adamts13* showed a predominant parietal association with cranial vault shape while being downregulated in the parietal mouse tissue. It is possible that the parietal tissue is more sensitive to alterations in corresponding protein levels. If so, genetic variation relating to gene regulation would be expected to have a more pronounced effect despite the lower overall transcriptional activity relative to the frontal tissue. Functional assays could test this idea.”

whereby we propose a possible and testable hypothesis.

- Candidate gene assignment relied on GREAT and FUMA analyses combined with manual curation. GTEx could provide further supporting evidence for a link to gene expression in e.g. skeletal tissues.

Author response:

We have now used GTEx data of 22 tissues (Supplementary Table 10) relevant for the cranial vault to look for eQTLs co-locating with our GWAS hits. Specifically, we looked at all protein coding genes within 1 Mb of our GWAS lead SNPs and performed Bayesian colocalization with the R package ‘coloc’¹³. Each gene with an eQTL overlapping with a GWAS loci with > 0.7 posterior probability was included as a candidate gene at that locus. All eQTLs overlapping with any GWAS locus with > 0.5 posterior probability are listed in Supplementary Table 11 with their corresponding tissue.

Additionally, we now moved the candidate genes per locus to Supplementary Table 1 and indicated for each candidate gene the source of its annotation (FUMA, GREAT, closest TSS, literature, and GTEx eQTL) to provide extra clarity to the reader.

Minor comments:

- Line 165, change ‘genome-wide significant’ to ‘genome-wide significance’.

Author response:

We thank the reviewer for noticing this typo. This was adapted accordingly.

- It would be useful to expand supplementary table 1 to include the coordinates of the 30 genome-wide significant loci identified after merging.

Author response:

At the moment, supplementary table 1 already contains the lead SNPs after merging the 15 segments. We have now also included in which segment the association was most significant.

- Supplementary table 4 is uninterpretable due to the large size. It should be provided as a spreadsheet, perhaps with an additional tab that limits the results to only the significant expression differences.

Author response:

We have now limited supplementary table 4 to our set of GWAS candidate genes only.

- “Other candidate genes were either not differentially expressed, had fewer than one gene count, or were not expressed at E15.5.”. Does ‘fewer than one gene count’ refer to the entire library or e.g. fewer than one count-per-million reads? Also, what is the threshold used to differentiate between

expressed and non-expressed genes?

Author response:

“Fewer than one gene count” is referring to the unnormalized raw count that was performed by the summarizeOverlaps function (vignette provided here for reference: <https://bioconductor.org/packages/devel/bioc/vignettes/GenomicAlignments/inst/doc/summarizeOverlaps.pdf>) to provide raw counts to Deseq2. For this setup, if there were fewer than 1 raw hit counted for any of the 6 grouped samples sequenced, the gene was removed from the analysis to speedup downstream computational analysis. We clarified this in the text. “Not expressed at E15.5” in this context means the same thing and refers to the same filtering. We removed this wording to avoid confusion.

The method used to calculate differential expression are referenced in (Love, Huber, and Anders 2014). Ultimately, the package estimates size factors (controls for differences in library size), calculates gene-wise dispersions, shrinks the estimates to more accurately model the read counts, and fits them to a gamma-Poisson distribution. A Wald test (taking the shrunken estimate of the logarithmic fold change divided by its standard error) is performed resulting in a Wald test p-value and adjusted for multiple testing using a Benjamini-Hochberg correction. An adjusted p-value of 0.05 was used as the cutoff to determine whether a gene was differentially expressed or not.

1. Gietzen, T. *et al.* A method for automatic forensic facial reconstruction based on dense statistics of soft tissue thickness. *PLOS ONE* **14**, e0210257 (2019).
2. J.C. Kolar, E. M. S. *raniofacial Anthropometry: Practical Measurement of the Head and Face for Clinical, Surgical and Research Use*. vol. 1997 (Springfield, IL: Charles C. Thomas).
3. White, J. D. *et al.* Insights into the genetic architecture of the human face. *Nat. Genet.* **53**, 45–53 (2021).
4. Willer, C. J., Li, Y. & Abecasis, G. R. METAL: fast and efficient meta-analysis of genomewide association scans. *Bioinformatics* **26**, 2190–2191 (2010).
5. Poole, W., Gibbs, D. L., Shmulevich, I., Bernard, B. & Knijnenburg, T. A. Combining dependent P-values with an empirical adaptation of Brown’s method. *Bioinformatics* **32**, i430–i436 (2016).
6. Cichonska, A. *et al.* metaCCA: summary statistics-based multivariate meta-analysis of genome-wide association studies using canonical correlation analysis. *Bioinformatics* **32**, 1981–1989 (2016).
7. Turley, P. *et al.* Multi-trait analysis of genome-wide association summary statistics using MTAG. *Nat. Genet.* **50**, 229–237 (2018).
8. Liu, Y. & Xie, J. Cauchy combination test: a powerful test with analytic p-value calculation under arbitrary dependency structures. *J. Am. Stat. Assoc.* **115**, 393–402 (2020).
9. Xiong, Z. *et al.* Combining genome-wide association studies highlight novel loci involved in human facial variation. *Nat. Commun.* **13**, 7832 (2022).
10. Li, J. & Ji, L. Adjusting multiple testing in multilocus analyses using the eigenvalues of a correlation matrix. *Heredity* **95**, 221–227 (2005).
11. Galwey, N. W. A new measure of the effective number of tests, a practical tool for comparing families of non-independent significance tests. *Genet. Epidemiol.* **33**, 559–568 (2009).
12. Li, M.-X., Gui, H.-S., Kwan, J. S. H. & Sham, P. C. GATES: A Rapid and Powerful Gene-Based Association Test Using Extended Simes Procedure. *Am. J. Hum. Genet.* **88**, 283–293 (2011).
13. Giambartolomei, C. *et al.* Bayesian Test for Colocalisation between Pairs of Genetic Association Studies Using Summary Statistics. *PLOS Genet.* **10**, e1004383 (2014).

REVIEWER COMMENTS

Reviewer #1 (Remarks to the Author):

The authors have addressed all of my concerns and suggestions. The updated figures and tables are appropriate and well-organized. I have no more comments on this version of the manuscript.

Reviewer #2 (Remarks to the Author):

I am not satisfied with the limited way the authors revised their manuscript concerning most of my previous comments. I can see in the authors extensive responses to my previous comments, but also to those of the other reviewers, that the authors prefer to argue and provide lots of text and other information in figures and tables in their rebuttal letter to support their views in the initially submitted manuscript, while use very little of all this to revise their manuscript accordingly. This is not the way how reviewers' comments should be treated by authors as far as I see it, unless a reviewer gets something completely wrong, which is not the case here as can be seen by the extensive replies the authors provide in their rebuttal letter. The reviewers identified issues that likely other readers will also see as problems later on. Replying in their rebuttal letter without revising their manuscript accordingly does not help solving these problems. Therefore, I encourage the authors to include their replies to my previous comments in their revised manuscript. If they decide to keep ignoring my previous comments in their revised manuscript, I am out of this process and it is up to the editor to make a decision without me as external reviewer.

Reviewer #3 (Remarks to the Author):

The authors have addressed all my prior comments satisfactorily and I have no further additions.

Reviewer #2 (Remarks to the Author):

I am not satisfied with the limited way the authors revised their manuscript concerning most of my previous comments. I can see in the authors extensive responses to my previous comments, but also to those of the other reviewers, that the authors prefer to argue and provide lots of text and other information in figures and tables in their rebuttal letter to support their views in the initially submitted manuscript, while use very little of all this to revise their manuscript accordingly. This is not the way how reviewers' comments should be treated by authors as far as I see it, unless a reviewer gets something completely wrong, which is not the case here as can be seen by the extensive replies the authors provide in their rebuttal letter. The reviewers identified issues that likely other readers will also see as problems later on. Replying in their rebuttal letter without revising their manuscript accordingly does not help solving these problems. Therefore, I encourage the authors to include their replies to my previous comments in their revised manuscript. If they decide to keep ignoring my previous comments in their revised manuscript, I am out of this process and it is up to the editor to make a decision without me as external reviewer.

Authors' response:

While it is hard to know what remains to be addressed, we did implement several additional changes to further improve the manuscript.

Specifically, we included an additional paragraph in the discussion which rephrases parts of our initial answer to the Reviewer's comment 3 as:

"As we have shown previously with facial surface shape and brain shape, the global-to-local segmentation approach used here yielded replicable associations that would otherwise have been missed by limiting to a global definition of the vault. While few additional hits were detected for the 14 segments that were not the global cranial vault, this could not have been known a priori given the poor understanding of normal-range cranial vault shape. Hence, we retained the segmentation approach to show that SNPs affecting vault shape mostly do so on a global level. Still, certain loci showed significant effects on local segments only, whereby the association strength decreased gradually going from a local to a global level, though still with some degree of association with the global cranial vault. This is exactly what one should expect if the true effect of a locus is indeed local, i.e., that considering a larger set of vertices increases the total variance in the phenotype while the variance explained by the genotype does not increase with the same extent, thereby resulting in a statistically weaker association. One example is rs3936018 near TBX15, which was associated specifically with the frontal region, and previously identified to be associated with forehead morphology. Another example is rs11609649 near ALX1, which also showed association locally with the frontal region, and not with the posterior regions, matching in situ expression patterns in mouse embryos. While for those loci, we observed effects in the regions where they were expected, it is interesting to see the lack of any significant association in other regions of the vault, thereby affirming their localized roles."

Additionally, we included Supplementary Methods 1 to provide more detail regarding the methodology used to address the Reviewer's comment 4.

Below is an overview of the initial numbered comments and responses. All changes to the manuscript resulting from those comments are summarized under each comment.

Initial comments by Reviewer #2

Reviewer comments (black) are followed by a summary of the comments; a list of how each point was addressed in the manuscript; and the original author responses. Text in blue relates to the initial review; while purple text documents the additional changes made based on the latest comment by Reviewer 2.

Reviewer #2 (Remarks to the Author):

The authors conducted a multi-ancestral GWAS together with a European replication study on human outer head surface and identified 30 genetic loci of which they replicated 20. While this reviewer appreciates the authors' efforts in unveiling the genetic basis of human head shape variation, there are several issues that make it difficult to know how novel and how reliable the reported findings are.

(each comment starts on a new page)

1) As phenotype basis for their GWAS, the authors used the outer surface of the head obtained from MRI, but report all their GWAS results for the cranial vault. They defend their approach by stating that the outer head surface represents a proxy for the cranial vault, but do not present evidence for this. What is the empirical evidence for this assumption? Why do they claim their results in connection with a proxy phenotype and not the phenotype they measured? This must be elaborated so that the reader can put the reported genetic findings in the right phenotypic perspective.

Author interpretation of the reviewers comments:

- a) The outer head surface is used to measure cranial vault shape variation, why not report results for “outer head surface”?
- b) If using outer head surface to capture the morphology of the underlying bones, what is the empirical evidence for this?
- c) In order for the reader to put findings in the right phenotypic perspective, it needs to be clear that the phenotype analyzed is the outer head surface.

Summary of revisions to the manuscript:

To address point b) we changed:

“Due to their close correspondence, the outer soft-tissue vault surface served as a proxy for the underlying cranial bones.”

To:

“Since the outer soft-tissue layer in this region is thin and uniform, shape variation associated with the neurocranial bones is well captured by our phenotypic definition.”

where we also cite Gietzen et al.

To avoid confusion in relation to point c), we changed:

“The cranial vault – the globular portion of the skull comprised of flat, plate-like bones that surrounds and protects the brain – shows considerable size and shape variation within and among human populations.”

To:

“The cranial vault – the globular portion of the head, shaped by flat, plate-like bones that surround and protect the brain – shows considerable size and shape variation within and among human populations.”

Author response to the reviewer:

Craniofacial malformations such as craniosynostosis demonstrate how variations in skull shape are clearly visible through the outer head surface. This should be no surprise since the soft tissue thickness surrounding the neurocranium is thin and does not vary much in thickness as illustrated in Figure 3 (<https://doi.org/10.1371/journal.pone.0210257.g003>) from Gietzen et al.¹. Therefore, it is possible to measure shape variation associated with the neurocranial bones through the outer head surface. This is reflected in our findings, which clearly demonstrate the skeletal roles of genes at our identified loci.

Fig 3. From Gietzen et al. 2019 – A method for automatic forensic facial reconstruction based on dense statistics of soft tissue thickness – Statistic of the FSTT on a mean skull. Mean and standard deviation of FSTT computed from the 43 CT scans. (<https://doi.org/10.1371/journal.pone.0210257.g003>)

Furthermore, in practice, measures of the boney cranial vault are routinely taken on the soft-tissue envelope. For example, in traditional anthropometry, measures of cranial vault shape are obtained by applying calipers to soft-tissue landmarks². In fact, these landmarks (glabella, opisthocranium, eurion) have the same names on the skull and immediately overlying soft tissue due to their close correspondence.

Because the vault refers to a broad anatomical region which may include the skull alone or the skull plus its soft tissue envelope, a precise phenotypic definition is required and given to avoid confusing the reader. We had previously already done so in the first sentence of the results section and in Extended Data Figure 1: “Cranial vault shape, herein defined as the outer head surface encompassing the supraorbital ridge and extending towards the occipital bone, was extracted from structural MRIs (Extended Data Fig 1; Methods)”

To further clarify that a regional definition for the cranial vault is used, we also modified the first sentence of the introduction to read:

“The cranial vault – the globular portion of the head, shaped by flat, plate-like bones that surround and protect the brain (...)”

Furthermore, we avoid using the term ‘proxy’ altogether, and changed:

“Due to their close correspondence, the outer soft-tissue vault surface served as a proxy for the underlying cranial bones.”

To:

“Since the outer soft-tissue layer in this region is thin and uniform, shape variation associated with the neurocranial bones is well captured by our phenotypic definition.”

2) Performing multi-ancestry GWAS is much less straightforward than single-ancestry GWAS and comes with several issues that can lead to false positive findings. This reviewer wonders how meaningful the multi-ancestry GWAS is in this case, where the majority of the discovery samples and all the replication samples are of single ancestry i.e., European and only a relatively small proportion of the discovery samples is of non-European ancestry. How do these limited non-European samples impact on the genetic findings? It is important to see the full results based on European-only samples and compare them with those from the multi-ancestry samples to get a practical idea how the intrinsic problems of multi-ancestry GWAS impacted on the presented results. The authors very briefly mention a comparison between multi-ancestry discovery and European-only discovery, but it seems this analysis was focussed on outcomes of the multi-ancestry GWAS and in any case was not presented with the needed level of detail. Both analyses, European-only and combined, should be reported and the differences critically discussed.

Author interpretation of the reviewer's comment:

- a) How do the limited non-European ancestry samples impact genetic findings?
- b) It is important to see the full European-only GWAS results.
- c) The European-only and multi-ancestry GWAS results need to be discussed more elaborately and in a way that does not focus solely on the multi-ancestry results.

Summary of revisions to the manuscript:

To address points a) and b), Supplementary Data 2 was added, listing all overlapping loci between the European-only, and Multi-ancestry GWAS. For each locus overlap, a figure is provided showing 1) the shape effects associated with the locus in both GWASs; 2) the P-value profile across the 15 hierarchical segments; and 3) LocusZoom plots and a LocusCompare plot to show the signal co-localization.

To address point b) and c), Supplementary Figure 8 was completely overhauled and now shows a Miami plot comparing both GWASs. This plot clearly indicates 1) which loci were shared; 2) which loci were unique and to which GWAS; and 3) in which GWAS the signal was the strongest.

To address point a) and c), results from both GWASs were analyzed and compared more elaborately in the results section of the manuscript. Initially the paragraph read:

*"We next performed a sensitivity analysis to evaluate the discovery of genome-wide loci when including or excluding non-European-ancestry GWAS cohort subjects. For each of the 30 genome-wide lead SNPs from the main GWAS ($n = 6,772$), we selected the most significant SNP from the European-only GWAS ($n = 4,198$) within 250 kb and in LD ($r^2 > 0.2$) to represent the same locus. In total 25 out of 30 (83.3%) loci reached a lower P-value in the multi-ancestry GWAS. Compared to the European-only GWAS, we identified 8 additional genomic loci at genome-wide significance ($P < 5e-8$) (Extended Data Fig 7) while only a single additional locus (*rs563186113*) reached genome-wide significance ($P = 4.10e-8$) in the European-only GWAS exclusively. The overall lower P-values and net gain of 7 additional genomic loci demonstrate the increased sensitivity resulting from the inclusion of non-European-ancestry participants."*

And was then changed into:

*"We next performed a sensitivity analysis to evaluate the discovery of genome-wide loci when including or excluding the 2,504 non-European-ancestry GWAS cohort subjects. To this end, a locus was considered to be overlapping between both GWASs if the corresponding lead SNPs were within 250 kb, resulting in a joint set of 31 unique loci (Supplementary Figure 8). Among these loci, 22 were shared, of which 17 (17/22; 77.3%) reached lower P-values in the multi-ancestry GWAS (Supplementary Figure 8). Moreover, 8 loci were unique to the multi-ancestry GWAS, and one locus (*rs563186113*) was unique to the European-only GWAS. Nonetheless, all those 9 loci still showed some degree of association in the other GWAS. Overall, the multi-ancestry GWAS thus achieved a*

higher discovery rate, identifying 7 more loci, and reaching lower P-values in 25 of the 31 loci (80.6%). Furthermore, shape effects and segment-wise association profiles were highly concordant between both GWASs at each of the 31 loci (Supplementary Data 2). At 21 loci (21/31; 67.7%), the same segment generated the lowest P-value, and at 5 more loci (5/31; 16.1%) the lowest P-values were generated in segments that were directly related. Strong co-localization was observed for the genomic signals in both GWASs as shown by LocusZoom and LocusCompare plots in Supplementary Data 2. Altogether, the analyses conducted in this section extensively demonstrated which aspects of the genetic findings were influenced by the inclusion of non-European individuals.”

Author response to the reviewer:

The ABCD cohort is an ancestrally diverse cohort with only 5,746 out of 11,099 (51.8%) European participants based on the ancestry inference described in the manuscript. The other 48.2% of participants were mostly admixed, had very little Asian ancestry, and could not be further stratified into any substantial parental populations. Therefore, the aim of this work has always been to conduct a multi-ancestry study as opposed to omitting almost half of the cohort in a Euro-centric study. While the literature is rapidly expanding with regards to multi-ancestry meta-analysis in stratified cohorts, literature on joint multi-ancestry and admixed GWAS remains relatively scarce. Thus, as a GWAS in an ancestrally complex US sample our study represents a valuable contribution to the existing literature.

Upon initial submission, our manuscript described how the inclusion of 2,504 non-European samples in a cohort of 6,772 US participants

- yielded stronger GWAS signals at the majority of loci shared with a European-only GWAS;
- yielded 8 additional genome-wide significant GWAS signals, while losing one relative to the European-only GWAS;
- reduced statistical power for high Fst SNPs;
- resulted in an enrichment of shared SNPs across ancestries;
- and yielded consistent effect sizes across ancestries.

We have now expanded the multi-ancestry versus European GWAS comparison. In a supplementary document we show that for each locus the latent shapes and association profile (i.e., the pattern of P-values across hierarchical segments) is highly concordant between both GWASs. Additionally, we show colocalization of the signal at each locus using LocusZoom and LocusCompare plots.

Furthermore, we updated the infographic that compared signal strength in both GWASs to be less focused on the multi-ancestry GWAS and better show which loci were shared or not. In conclusion, we have extensively demonstrated how the inclusion of non-European individuals affected genetic findings. This paragraph was extensively reworked in the text and now reads:

“We next performed a sensitivity analysis to evaluate the discovery of genome-wide loci when including or excluding the 2,504 non-European-ancestry GWAS cohort subjects. To this end, a locus was considered to be overlapping between both GWASs if the corresponding lead SNPs were within 250 kb, resulting in a joint set of 31 unique loci (Supplementary Figure 8). Among these loci, 22 were shared, of which 17 (17/22; 77.3%) reached lower P-values in the multi-ancestry GWAS (Supplementary Figure 8). Moreover, 8 loci were unique to the multi-ancestry GWAS, and one locus (rs563186113) was unique to the European-only GWAS. Nonetheless, all those 9 loci still showed some degree of association in the other GWAS. Overall, the multi-ancestry GWAS thus achieved a higher discovery rate, identifying 7 more loci, and reaching lower P-values in 25 of the 31 loci (80.6%). Furthermore, shape effects and segment-wise association profiles were highly concordant between both GWASs at each of the 31 loci (Supplementary Information 2). At 21 loci (21/31; 67.7%), the same segment generated the lowest P-value, and at 5 more loci (5/31; 16.1%) the lowest P-values were generated in segments that were directly related. Strong co-localization was observed for the genomic signals in both GWASs as shown by LocusZoom and LocusCompare plots in Supplementary Information 2. Altogether, the analyses conducted in this section extensively

demonstrated which aspects of the genetic findings were influenced by the inclusion of non-European individuals”

3) The segmentation approach the authors applied, which they developed previously regarding the face, results in several segments of the outer head surface for which the biological relevance appears largely questionable if not being absent at all. This reviewer cannot see that the segmentation approach was done with a biological hypothesis in mind, which in principle should not be done when aiming to study a biological phenomenon. Notably, most of the GWAS findings they reported were obtained for “global vault shape”, i.e., the total outer head surface as measured without any segmentation. Only very few hits were seen for any of the 14 segments generated from the total surface with the segmentation approach. This reviewer largely questions the use of the segmentation approach for this phenotype and remains very sceptical about the true positive nature of the genetic loci reported solely for segments not overlapping with those identified with total shape. Moreover, doing GWAS on multiple phenotypes comes for the price of having to consider multiple phenotypes and their correlation in the analysis and interpretation, which the authors mostly ignored (see next point). Perhaps a GWAS on the single phenotype of total outer head surface would be more powerful and deliver more trustworthy results?

Author interpretation of the reviewer’s comment:

- a) The biological underpinnings of the segmentation approach are not obvious.
- b) The associations with smaller segments may be false positives.
- c) It could be beneficial to only report on the global vault GWAS, and omit the segmentation.

Summary of revisions to the manuscript:

To address a), we performed an additional analysis to investigate how genetic effects vary in magnitude across the cranial vault and added Supplementary Figure 3. Based on this analysis, it appears that the data-driven segmentation captures units of shape variation as local foci of deformation rather than individual bones. This was reported on in results as:

“To demonstrate that our data-driven segmentation approach is capturing genetically salient aspects of vault shape variation, we examined the effects of discovered SNPs per vertex on the entire dense 3D mesh. Based on the 21 lead SNPs that were significantly ($P < 5e-8$) associated with global vault shape, we calculated deformation hotspots (Supplementary Fig. 3), which we defined as local regions of the cranial vault where effects attained a greater magnitude relative to their immediate surroundings. Most of these hotspots were located along the midline, coinciding with sutures, and two hotspots were located laterally. These results show that without including any SNPs whose phenotypic associations were dependent on the segmentation approach, a pattern of deformation hotspots was obtained that aligns almost perfectly with our vault segmentation. This provides confirmation that our data-driven approach can capture biologically meaningful information and is capable of doing so with a fraction of the computational bandwidth required to perform vertex-wise analyses.”

And with corresponding methods reported on in the methods section.

To address b), we adapted the infographics in Supplementary Data 1 to more clearly show which associations and which segments were replicated in the UK Biobank sample, thereby clearly showing a good replication rate for smaller segments (taking in consideration the damage in the replication images).

Our line of reasoning regarding point c) and as expressed in the author response below, was added to the discussion as:

“As we have shown previously with facial surface shape and brain shape, the global-to-local segmentation approach used here yielded replicable associations that would otherwise have been missed by limiting to a global definition of the vault. While few additional hits were detected for the 14 segments that were not the global cranial vault, this could not have been known a priori given the poor understanding of normal-range cranial vault shape. Hence, we retained the segmentation

approach to show that SNPs affecting vault shape mostly do so on a global level. Still, certain loci showed significant effects on local segments only, whereby the association strength decreased gradually going from a local to a global level, though still with some degree of association with the global cranial vault. This is exactly what one should expect if the true effect of a locus is indeed local, i.e., that considering a larger set of vertices increases the total variance in the phenotype while the variance explained by the genotype does not increase with the same extent, thereby resulting in a statistically weaker association. One example is rs3936018 near TBX15, which was associated specifically with the frontal region, and previously identified to be associated with forehead morphology. Another example is rs11609649 near ALX1, which also showed association locally with the frontal region, and not with the posterior regions, matching *in situ* expression patterns in mouse embryos. While for those loci, we observed effects in the regions where they were expected, it is interesting to see the lack of any significant association in other regions of the vault, thereby affirming their localized roles."

Author response to the reviewer:

Since the aim of the study was to study cranial vault shape variation, we reasoned that shape variation could directly inform the segmentation, resulting in an unbiased and data-driven approach, such as used in earlier work on the face and brain. Additionally, the segmentation approach works in conjunction with PCA, i.e., grouping correlated vertices yields more efficient dimensionality reduction, aiding with association testing. This approach represents only one of many ways to segment the cranial vault. The resulting segment edges do not coincide with cranial sutures, however, that does not indicate a lack of biological meaning. Fundamentally, a data-driven segmentation will capture biological aspects of shape variation that are not immediately obvious to a single human observer.

To attempt to expose the underlying biological drivers of the segmentation, we examined the vertex-wise shape effects of all lead SNPs with genome-wide significant effects on global cranial vault shape, i.e., those SNPs whose discovery was not dependent on the segmentation approach. Specifically, we looked at the regions on the cranial vault where those SNPs effects were the largest by looking at the 95th percentile of absolute shape deformation associated with each vertex. This revealed deformation hotspots, many of which located on the midline, and thus clearly coinciding with sutures. Interestingly, overlaying our data-driven segmentation revealed how the deformation hotspots get more isolated and focused on within each hierarchical layer of segmentation. Strikingly, the smallest segments were centered around specific hotspots. Hence, we show how our data-driven segmentation has plausible biological underpinnings and is clearly influenced by suture-associated shape variation. Supplementary Figure 3 shows these segments and hotspots.

As we have shown previously with facial surface shape and brain shape, the global-to-local segmentation approach used here yields replicable associations what would otherwise have been missed by limiting to a global definition of the vault. While it is true that few additional hits were seen for the other 14 segments (all segments, excluding the full cranial vault), this could not be known a priori given the poor understanding of normal-range cranial vault shape. Hence, we retained the segmentation approach to show that SNPs affecting vault shape mostly do so on a global level, which is a result in itself. Still, certain loci showed strong effects on local segments only. Examples include rs3936018, near *TBX15* associated specifically with the frontal region ($P = 4.10e-19$), and previously identified to be associated with forehead morphology³. Another example is rs11609649 near *ALX1*, which also showed association locally with the frontal region ($P = 8.24e-10$), and not with the posterior regions, matching *in situ* expression patterns in mouse embryos (<https://www.informatics.jax.org/assay/MGI:3508780>). While for those loci, we observed effects in the regions where they were expected, it is interesting to see the lack of significant association in other regions of the vault and affirming their localized roles. Moreover, we identified a signal near

SHH, which was only locally associated with cranial vault shape, has a well-established role in craniofacial development, and was replicated in the UKBB.

Regarding the remark that loci with local effects still show some association for the global cranial vault, Supplementary Information 1 clearly shows that the global shape effects of those loci are consistent with the local shape effects. Also note that the association profile across the segments is not random, often showing a gradient of association strength from local to global segments. This is exactly what one should expect if the true effect of a locus is indeed local, i.e., that considering a larger set of vertices increases the total variance in the phenotype while the variance explained by the genotype does not increase with the same extent, thereby resulting in a statistically weaker association. We have updated the infographics in Supplementary Information 1 to better show this. Additionally, we indicated which segments were replicated in the UKBB.

4) The authors performed 15 separate GWASs for the 15 phenotypes they used from applying their segmentation approach. It seems they did not do a meta-analysis on the outcomes of these 15 GWASs, which would be the normal way to do. Since meta-analysis is a well-established approach, the question appears why the authors deviated from this? If the authors were to stick with the multiple phenotypes they generated with their segmentation approach despite the criticism expressed under point 3, meta-analysis should be conducted and significant study results should only be concluded from the meta-analysis outcomes. Also, it can be assumed that there are correlations between these 15 phenotypes used, which, in case the authors keep using them, should be tested empirically and such correlations, if identified, should be taken into account in the GWAS in an appropriate statistical way.

Author interpretation of the reviewer’s comment:

- a) The 15 GWASs were not meta-analyzed, even though this would be the ‘normal’ thing to do.
- b) Correlations between the phenotypes should be tested empirically and taken into account.

Summary of revisions to the manuscript:

While point b) was already addressed by the original manuscript, additional calculations were performed to show robustness and were reported in the text as:
“Each permuted genotype was tested for association with the 15 cranial vault segments and the lowest P-value was retained. We then divided 0.05 by the 5th percentile of the resulting 10,000 P-values to estimate the number of effective GWAS runs performed. This was repeated for 500 random SNPs, resulting in an average effective number of phenotypes of 11.44 (SD: 0.56). The effective number of phenotypes was additionally estimated following the same approach using the 30 lead SNP as well as using three eigenvalue-based methods^{157–159} on the 15 x 15 segment-segment correlation matrix as estimated using either genome-wide P-values, chi squared statistics, or genomic Pearson correlations, yielding 10 additional estimates in the range of 8.52 – 11.28. We then opted for the former, most conservative estimate, and obtained a subsequent ‘study-wide’ significance threshold of $P < 4.37e-9$ (i.e., $5e-8 / 11.44$).”

The methodology used to address point b) is now described in detail in Supplementary Methods 1.

Author response to the reviewer:

Many GWAS meta-analyses assume that there is no sample overlap between GWASs, such as Fisher’s P, Stouffers Z, and inverse-variance weighted (IVW) meta-analysis (which additionally assumes the same effect size across traits). This assumption is violated in our study, as samples fully overlap between each of the 15 GWASs. Furthermore, most other approaches require (signed) effect size estimated and their standard errors (e.g., METAL, MTAG, metaCCA, C-GWAS) which are not yielded by our multivariate regression using CCA. Hence, while meta-analysis of GWAS is common practice, most of the current implementations are simply not compatible with studies like ours where there is complete sample overlap and no signed effect size estimates (and their standard errors) are available. We therefore went with an approach that makes no assumptions on sample overlap, requires only P-values, and has been applied before in similar studies (Naqvi et al., White et al.). An overview of meta-analysis methods is given in the table below.

Method	Citation	Assumptions	Input	Why this method is not applicable to our cranial vault GWAS
Fisher’s P / Stouffers Z		No sample overlap	P-values	Samples overlap fully between segments.

Inverse variance weighted / Fixed effect meta-analysis		No sample overlap, beta constant across traits	Signed Z-scores (or signed betas), standard errors on beta	Samples overlap fully between segments. CCA does not yield z-scores, nor betas. Effects likely differ across segments.
minP	Tippett (1931)	Samples can overlap	P-values, individual level data (to calculate N_{eff})	Applicable!
METAL	Willer (2010) ⁴	Samples can overlap	Signed Z-scores or beta + SE (based on IVW)	CCA does not yield signed betas
Empirical Brown's Method / Kost's Mehtod	Poole (2016) ⁵	Samples can overlap	P-values, individual level univariate traits (can be adapted)	Applicable!
metaCCA	Cichonska (2016) ⁶	Samples can overlap	Beta + SE	CCA does not yield betas
MTAG	Turley (2018) ⁷	Samples can overlap (Relies on bivariate LD-score regression to estimate sample overlap)	Signed Z-scores or beta + SE (based on IVW)	Relies on LD-score regression: - No stadardized implementation for multivariate test statistics - Does not work in multi-ancestry cohort.
Cauchy test	Liu (2020) ⁸	Samples can overlap	Pvalues	Applicable!
C-GWAS	Xiong (2022) ⁹	Samples can overlap	P-values, signed betas (based on IVW)	CCA does not yield signed betas

To combine our GWAS results across multiple segments, we take for each SNP the minimal P-value across the 15 segments (minP). Since correlation between the segments is expected (due to e.g., the hierarchical nature of segments), a Bonferroni correction based on 15 independent tests would be overly stringent. For a more accurate adjustment, we estimated the number of effective independent tests per SNP following Kanai (2016) and ran 10,000 genotype-phenotype associations under the null by permutation testing in a way that preserved the correlational structure of our phenotypes. This was repeated for 500 random SNPs and yielding an estimate of 11.44 (± 0.55) independent traits.

To illustrate the robustness of this estimate, we have additionally estimated the number of effective traits using several other approaches:

First, we repeated the permutation testing for our 30 lead SNPs, yielding an estimate for the number of effective traits of 11.28 (± 0.50).

Second, we applied several methods to estimate the number of effective traits from a trait-correlation matrix. We used three methods: 1) Li & Ji (2005)¹⁰, 2) Galwey (2009)¹¹, and 3) Li (2011)¹² on three different correlation matrices: 1) based on the P-values from the GWAS, 2) based on the Chi-squared statistics from the GWAS, and 3) based on the Spearman genetic correlation matrix (methods). All three correlation matrices yielded highly concordant results across methods.

Method	P-values	Chi-squares	Genetic correlation
--------	----------	-------------	---------------------

Li & Ji (2005)	11.00	10.00	10.00
Galwey (2009)	11.27	10.84	10.98
Li (2011)	8.87	8.52	8.57

While these estimates mostly agree on a number of effective traits, the most conservative estimate of 11.44 was obtained based on 10,000 permutations of 500 random SNPs. We therefore adjusted the genome-wide significance threshold (instead of the P-values themselves) using Bonferroni correction based on 11.44 effective traits, resulting in a study-wide threshold of $4.37e-9$ (i.e., $5e-8 / 11.44$).

In short, we tried several approaches to estimate the number of effective traits, thereby taking into account the correlational structure between the phenotypes. We showed robustness of our estimates and used the most conservative estimate to adjust the significance threshold.

5) As far as this reviewer can see, the authors left out describing which of the genetic loci they identified were discovered for the first time for their phenotype(s), therefore representing new knowledge, and which were already identified in previous studies and represent confirmation of previously established knowledge. It is unexpected that the authors did not look into this given the novelty focus of scientific journals. Why did they not describe this? Given that this is not the first genetic study on head shape, it should be clearly described how many and which loci represent novel knowledge and how many and which confirm previous knowledge.

Author interpretation of the reviewer's comment:

a) Novel and known loci should be announced as such.

Summary of revisions to the manuscript:

To address a), the following sentence was added to the results section:

"Among these signals, only the locus near HMGA2 has been previously identified in GWAS on cranial vault dimensions."

and Figure 5 was updated to show overlapping signals with forehead morphology in White et al. (2021).

Author response to the reviewer:

Only *HMGA2* was previously identified in two separate GWASs on head circumference. None of the other loci were previously identified in GWAS on cranial vault dimensions (head circumference, maximum cranial length, maximum cranial length, and cephalic index). This has now been indicated in the text:

Among these signals, only the locus near HMGA2 has been previously identified in GWAS on cranial vault dimensions.

As already mentioned in the text, several loci overlap with the facial GWAS of White et al. 2021. These loci and their associated segments are shown in the updated Figure 5.

References

1. Gietzen, T. *et al.* A method for automatic forensic facial reconstruction based on dense statistics of soft tissue thickness. *PLOS ONE* **14**, e0210257 (2019).
2. J.C. Kolar, E. M. S. *raniofacial Anthropometry: Practical Measurement of the Head and Face for Clinical, Surgical and Research Use.* vol. 1997 (Springfield, IL: Charles C. Thomas).
3. White, J. D. *et al.* Insights into the genetic architecture of the human face. *Nat. Genet.* **53**, 45–53 (2021).
4. Willer, C. J., Li, Y. & Abecasis, G. R. METAL: fast and efficient meta-analysis of genomewide association scans. *Bioinformatics* **26**, 2190–2191 (2010).
5. Poole, W., Gibbs, D. L., Shmulevich, I., Bernard, B. & Knijnenburg, T. A. Combining dependent P-values with an empirical adaptation of Brown's method. *Bioinformatics* **32**, i430–i436 (2016).
6. Cichonska, A. *et al.* metaCCA: summary statistics-based multivariate meta-analysis of genome-wide association studies using canonical correlation analysis. *Bioinformatics* **32**, 1981–1989 (2016).
7. Turley, P. *et al.* Multi-trait analysis of genome-wide association summary statistics using MTAG. *Nat. Genet.* **50**, 229–237 (2018).
8. Liu, Y. & Xie, J. Cauchy combination test: a powerful test with analytic p-value calculation under arbitrary dependency structures. *J. Am. Stat. Assoc.* **115**, 393–402 (2020).
9. Xiong, Z. *et al.* Combining genome-wide association studies highlight novel loci involved in human facial variation. *Nat. Commun.* **13**, 7832 (2022).
10. Li, J. & Ji, L. Adjusting multiple testing in multilocus analyses using the eigenvalues of a correlation matrix. *Heredity* **95**, 221–227 (2005).
11. Galwey, N. W. A new measure of the effective number of tests, a practical tool for comparing families of non-independent significance tests. *Genet. Epidemiol.* **33**, 559–568 (2009).
12. Li, M.-X., Gui, H.-S., Kwan, J. S. H. & Sham, P. C. GATES: A Rapid and Powerful Gene-Based Association Test Using Extended Simes Procedure. *Am. J. Hum. Genet.* **88**, 283–293 (2011).